# Moonlighting Arabidopsis molybdate transporter 2 family and GSH-complex formation facilitate molybdenum homeostasis

Jan-Niklas Weber [1,5], Rieke Minner-Meinen [1,5], Maria Behnecke[1], Rebekka Biedendieck [2], Veit G. Hänsch [3], Thomas W. Hercher[1], Christian Hertweck [3], Lena van den Hout[1], Lars Knüppel[1], Simon Sivov[1], Jutta Schulze[1], Ralf-R. Mendel[1], Robert Hänsch [1,4✉] & David Kaufholdt [1]

Molybdenum (Mo) as essential micronutrient for plants, acts as active component of molybdenum cofactor (Moco). Core metabolic processes like nitrate assimilation or abscisic-acid biosynthesis rely on Moco-dependent enzymes. Although a family of molybdate transport proteins (MOT1) is known to date in *Arabidopsis*, molybdate homeostasis remained unclear. Here we report a second family of molybdate transporters (MOT2) playing key roles in molybdate distribution and usage. KO phenotype-analyses, cellular and organ-specific localization, and connection to Moco-biosynthesis enzymes *via* protein-protein interaction suggest involvement in cellular import of molybdate in leaves and reproductive organs. Furthermore, we detected a glutathione-molybdate complex, which reveals how vacuolar storage is maintained. A putative Golgi S-adenosyl-methionine transport function was reported recently for the MOT2-family. Here, we propose a moonlighting function, since clear evidence of molybdate transport was found in a yeast-system. Our characterization of the MOT2-family and the detection of a glutathione-molybdate complex unveil the plant-wide way of molybdate.

[1] Institute of Plant Biology, Technische Universität Braunschweig, Humboldtstrasse 1, D-38106 Braunschweig, Germany. [2] Institute of Microbiology and Braunschweig Integrated Centre of Systems Biology, Technische Universität Braunschweig, Rebenring 56, D-38106 Braunschweig, Germany. [3] Department of Biomolecular Chemistry, Leibniz Institute for Natural Research and Infection Biology (HKI), Beutenbergstrasse 11a, Faculty of Biological Sciences, Friedrich Schiller University Jena, D-07743 Jena, Germany. [4] Center of Molecular Ecophysiology (CMEP), College of Resources and Environment, , Southwest University, Tiansheng Road No. 2, 400715 Chongqing, Beibei District, PR China. [5] These authors contributed equally: Jan-Niklas Weber, Rieke Minner-Meinen ✉email: r.haensch@tu-braunschweig.de

Moco-biosynthesis is crucial for the functionality of Moco-dependent enzymes (Moco-enzymes), key elements of basic metabolic pathways *in planta*[1]. The four-step process involves six enzymes and is conserved throughout all kingdoms of life[2]. Molybdate insertion into molybdopterin is maintained by molybdenum-insertase Cnx1 (Cofactor for nitrate reductase and xanthine dehydrogenase 1) acting in a multi-enzyme complex[3]. The Moco key user nitrate reductase (NR) catalyzes the first step of nitrate assimilation[4]. Moco-enzymes are, *inter alia*, involved in abscisic acid (ABA) biosynthesis by the abscisic aldehyde oxidase (AAO)[5] and sulfite detoxification by the sulfite oxidase[6].

Efficient absorption of molybdate ($MoO_4^{2-}$) and distribution from soil into cells is essential for Moco-biosynthesis in plants and is maintained by specialized molybdate transporter (MOT) proteins[7]. Two independent MOT families are thought to exist[8,9], however, only members of the MOT1-family have been characterized to date, playing distinct physiological roles in Moco-biosynthesis[7].

MOT1.1 (formerly known as MOT1[7]; AT2G25680) is a plasma membrane (PM) protein with a high-affinity molybdate transport activity (Km of 20 nM)[10] and is most prominently produced in roots[11]. Therefore, MOT1.1 functions as radicular molybdate importer from soil but is not involved in leaf cell import or in molybdate delivery to the Moco-biosynthesis complex[7]. The highly-related, tonoplast-localized MOT1.2 (formerly known as MOT2[7]; AT1G80310)[12] is the second member of the MOT1-family. It controls the release of stored molybdate from the vacuole by direct interaction with Cnx1[7].

Despite the reported functions of MOT1-family members, two mechanisms remained unclear to fully understand molybdate homeostasis in plants: (i) how molybdate enters leaf and seed cells and (ii) how it is imported into the vacuole. It is conceivable that members of the second MOT-family might maintain these tasks.

The independent and unrelated MOT2-family was first discovered in *Chlamydomonas reinhardtii*[8]. *Cr*MOT2 showed high-affinity molybdate uptake activity (Km of 550 nM) when heterologously produced in yeast. Orthologues of genes are present in most eukaryotes, including animals[8] and higher plants like *Oryza sativa*[13]. Even though the presence of three MOT2-family members in *Arabidopsis thaliana* (*A. thaliana*) were postulated[8,9], as to yet, a MOT function for this family has not been proven[9,14].

Recently, *Arabidopsis* MOT2-family was associated with an additional putative S-adenosyl methionine (SAM) Golgi import function for polysaccharide methylation, necessary for correct cell wall biosynthesis[14].

Here we report the identification of four members of the MOT2-family in *A. thaliana* and show their molybdate transport activity. PM-localization and interaction with molybdenum-insertase Cnx1 reveal their role as molybdate importers. Due to the additional function as Golgi-SAM importers, we postulate a moonlighting character. Our findings of global expression of one MOT2-family member and its importance for Moco key-user NR show its pivotal role in molybdate distribution and cellular import to directly supply Moco-biosynthesis. Remaining MOT2-family members are produced in flowers and a role in seed and pollen maturation is suggested. We detected a glutathione-molybdate complex, which shows how vacuolar storage of molybdate is functioning. Altogether, our findings outline the plant-wide way of molybdate from uptake, distribution, storage, and usage in Moco-biosynthesis.

## Results

### In silico identification of the MOT2 family in *A. thaliana*.
Two MOT-families are present in *Chlamydomonas reinhardtii* consisting of one member each: *Cr*MOT1[15] and *Cr*MOT2[8]. According to its homology to *Cr*MOT2 the protein *At*MOT2.1 of *A. thaliana*

encoded by gene AT4G27720[8] was identified. The encoded proteins *Cr*MOT2 and *At*MOT2.1 share an amino acid (aa) similarity of 71% and contain four highly conserved motifs[8] characteristic for MOT2 proteins (Fig. 1a). In a phylogenetic analysis, Huang et al.[9] identified two more *mot2*-related genes in *A. thaliana*: *Atmot2.2* (AT1G64650) and *Atmot2.3* (AT3G49310). Analyses of the encoded proteins revealed a sequence similarity of over 83% on aa level, showed their association to the major facilitator superfamily, and, most importantly, identified four MOT2 specific motifs (Fig. 1a). *mot2.2* encodes for two proteins: MOT2.2 A consists of 462 aa and its splice variant MOT2.2B which lacks the first 41 aa. MOT2.3 consists of 460 aa.

### Characterization of the MOT2-family as molybdate transporters.
Molybdate transport function of the MOT2-family was tested by an in vivo yeast-based growth assay. Wild type (WT) *Saccharomyces cerevisiae* lacks genes connected to molybdenum metabolism, including MOTs[16]. Fungal genes encoding the Moco-biosynthesis and NR derived from *Neurospora crassa* were integrated into the yeast genome (Table S1) following the approach of Perli et al.[17], and both Moco-biosynthesis as well as NR-activity were detected when co-expression of *mot* candidate genes was induced in growth media containing a physiological amount of 100 nM molybdate.

As next step, an in vivo-growth assay with the Moco-yeast strains was performed. NR not only reduces nitrate to nitrite but also chlorate to cytotoxic chlorite[18,19]. When co-expressed *mot* candidates are capable of active molybdate transport, Moco-yeasts become more sensitive towards chlorate due to functional Moco-biosynthesis which yields active NR capable of producing cytotoxic chlorite. Resulting growth inhibition can be quantified using a BioLector® growth analyzer.

Moco-yeasts expressing *mot1.1* as positive control (Fig. 1b) showed significantly impaired growth in presence of chlorate and molybdate when gene expression was induced by galactose. Analyses of *mot1.2* and all *mot2* candidates (Fig. 1c) revealed significant growth inhibition after galactose induction comparable to the positive control. Thus, induced expression of all tested *mot2* candidates caused a molybdate transport dependent growth inhibition based on the production of cytotoxic chlorite due to an increased NR-activity. Furthermore, an apparent order of molybdate transport capacity was observed (Fig. 1d). MOT1.1 and MOT1.2 showed the highest capacity, followed by MOT2.3, MOT2.2B, MOT2.2 A, and MOT2.1.

### Localization and topology studies of MOT2-family.
Intracellular localization of MOT2 was elucidated by fluorescence microscopy in *Nicotiana benthamiana* (*N. benthamiana*) mesophyll protoplasts. All N-terminally fused Venus-MOT2 constructs showed no or only weak signals. Consequently, only C-terminally fused MOT2-Venus constructs were analyzed. All MOT2-family members localized similarly as a thin layer encircling the protoplast at its periphery (Fig. 2). Co-expression with the cytosol marker eqFP611 (Fig. 2a1–d1) showed clear differentiation of the signals whereas co-localization with the PM-marker *At*PIP2A[20] (Fig. 2a2–d2) was observed indicating PM-localization of MOT2. In addition, fluorescence was detected in vesicle like structures. An additional localization experiment in *A. thaliana* seedlings verified PM-localization and showed co-localization with the *cis*-Golgi marker *Gm*Man-1[21] (Fig. S2).

Topology studies were performed to determine N- and C-terminus orientation using a split-GFP system in *N. benthamiana* leaf epidermis cells[22]. MOT2 proteins were tagged N- (Fig. 2e1 + 2, 2g1 + 2, 2i1 + 2 and 2k1 + 2) or C-terminally (Fig. 2f1 + 2, 2h1 + 2, 2j1 + 2 and 2l1 + 2) with the 11th β sheet of GFP (GFP11) and were co-expressed with either a

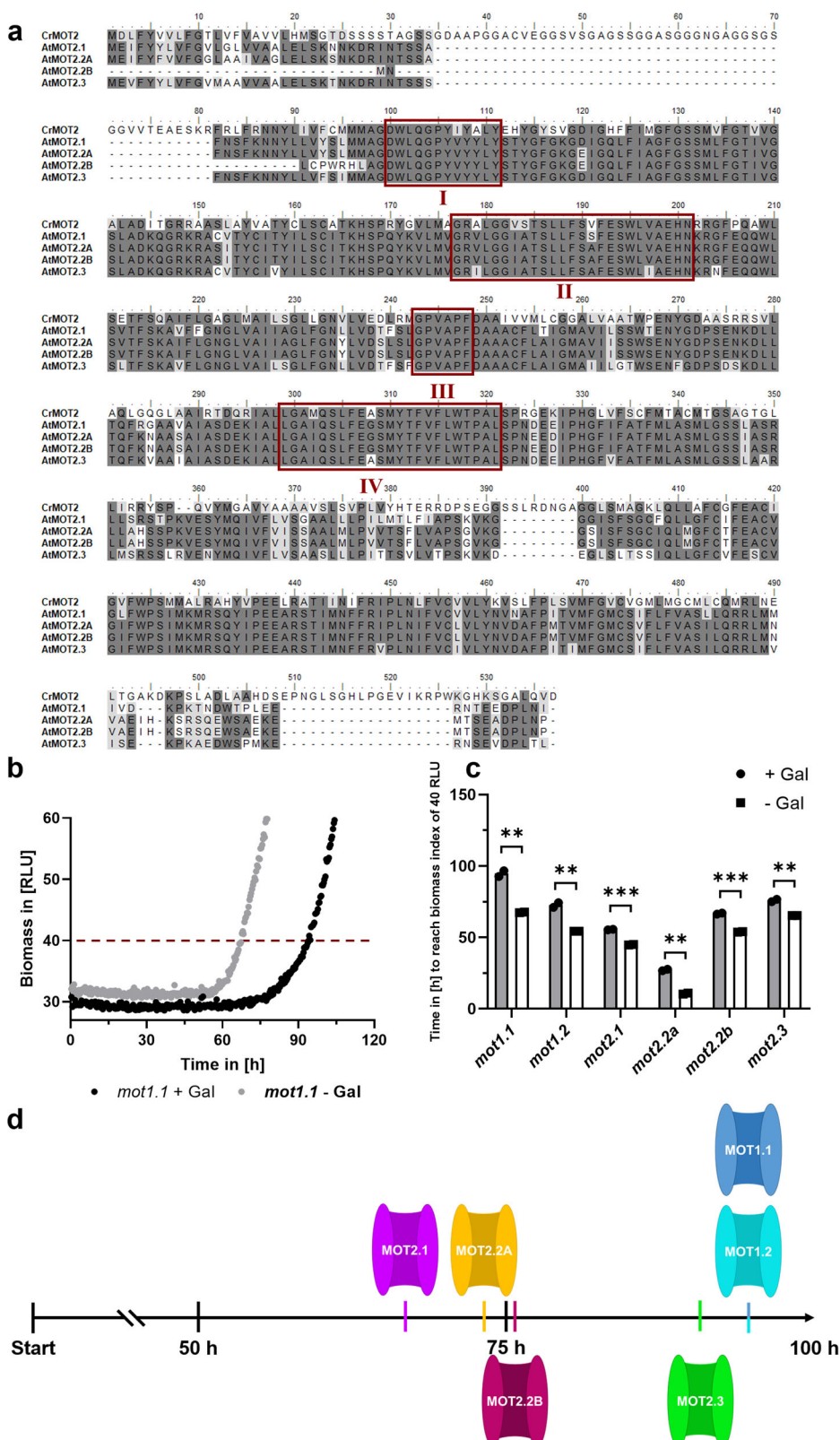

cytosolic GFP1-10 or with GFP1-10 tagged with an apoplast signal peptide (SP-GFP1-10). All MOT2-family members tagged N-terminally with GFP11 (GFP-11-MOT2) showed a fluorescence signal after reporter reconstitution with the apoplastic SP-GFP1-10 (Fig. 2e2, g2, i2 and k2). Fluorescence was also detected after all C-terminally GFP-11-tagged MOT2-family members (MOT2-GFP11) interacted with cytosolic GFP1-10 (Fig. 2f1, h1,

j1 and l1). Consequently, the N-terminus of all MOT2 proteins is oriented into the apoplast, whereas the C-terminus is oriented into the cytosol further underlining their PM-localization.

**Organ-specific expression pattern of *mot2* genes.** To assign a physiological role to MOT2-family members their spatial and temporal organ-specific expression patterns were studied.

**Fig. 1 MOT2-family sequence alignment and impact of MOT2 mediated molybdate import on *Saccharomyces cerevisiae*. a** ClustalW alignment of MOT2 sequences from *Chlamydomonas reinhardtii* (*Cr*) and *A. thaliana* (*At*). Identical aa are marked in dark grey, similar in light grey and different in white. Four highly conserved domains are marked with red boxes. **b** Representative growth curve of Moco-yeasts in molybdate-chlorate medium with induced (black; +Gal) and non-induced (grey; –Gal) *mot1.1* expression. Biomass index of 40 RLU is marked with a red dotted line. **c** Time in hours to reach a biomass index of 40 RLU of Moco-yeasts with induced (black; +Gal) and non-induced (white, –Gal) *mot* expression. Mean of two independent experiments is plotted and error bars depict standard deviation. Unpaired *T*-test was used to test for significance. **P ≤ 0.01; ***P ≤ 0.001. **d** Ranking according the time to reach biomass index of 40 RLU of Moco-yeasts with induced *mot* expression as indicated. Growth behaviour was normalized to chlorate-free controls. Quantitative data in Fig. 1c was additionally analyzed by two-way ANOVA and Sidak post hoc test for multiple comparison (Table S2).

Histochemical analysis of *mot2.1:gus* plants (Fig. 3a–e) showed signals in roots concentrated in vascular tissue. Expression was also detected in epidermis and vascular tissue of shoots and in developing flowers with strong signals in ovaries. Young leaves showed global expression reduced to main leaf venation with increasing age (Fig. 3b, c) as also quantified by fluorimetric assay (Fig.3f). Interestingly, the fluorimetric assay also revealed a significant increase of *mot2.1* expression under molybdate abundance in all organs except the flower. *mot2.2:gus* plants showed very distinct signals in pollen grains of mature flowers (Fig. 3g). Analysis of *mot2.3:gus* showed signals in ovaries (Fig. 3h). Availability of molybdate had no influence on expression patterns of *mot2.2:gus* and *mot2.3:gus*. Expression of *mot2.2* and *mot2.3* in the flower did not allow a fluorimetric analysis.

**Loss of MOT2-family members partially affects nitrate reductase activity**. Molybdate deprivation has a striking effect on plants and leads to reduced activity of Moco-enzymes resulting in retarded growth, leaf necrosis, dwarfism, and ultimately death[23]. Loss of MOTs can cause this phenotype under molybdate shortage, as demonstrated for members of the MOT1-family[7]. Phenotype of *Arabidopsis mot2* T-DNA knockout (KO) lines (Table 1) were analyzed to reveal whether the loss of MOT2-family members impacts vegetative growth and molybdate homeostasis. The interplay between members of the MOT1- and the MOT2-family in maintaining molybdate homeostasis was analyzed by characterizing KO lines with a higher degree in comparison to WT and a *mot1.1* single-KO, encoding the main root molybdate importer MOT1.1, regarding their impact on vegetative growth. A loss of the vacuolar exporter MOT1.2 showed no effects neither on the macroscopic nor the molecular phenotype as observed in a recent study[7]. Due to the organ-specific expression patterns of *mot2.2* and *mot2.3* (Fig. 3g, h; Fig. S3), both were excluded from higher degree KO analyses.

Interestingly, no notable alterations in plant survival, development (Fig. S4) or growth behavior were observed (Fig. 4a–c). As MOT2.1 is primarily present in root vascular tissue (Fig. 3a), a molybdate uptake assay was performed to test whether uptake is impaired in *mot2.1*-KO plants (Fig. 4d). Whereas molybdate uptake rate of the positive control *mot1.1*-KO was significantly decreased compared to WT, *mot2.1*-KO showed only a minor reduction. NR-activity is directly linked to Moco availability and, thus, an important indicator to show functionality of molybdate supply. Molybdate deprivation reduced NR-activity in the WT (Fig. 4e) significantly from 0.47 nmol · (h · mg$_{FW}$)$^{-1}$ by 32% to 0.32 nmol · (h · mg$_{FW}$)$^{-1}$. Comparable results were observed for *mot2.2*-KO and *mot2.3*-KO. Interestingly, *mot2.1*-KO showed a significant reduction of NR activity by 30 % under both, molybdate availability and deprivation conditions.

The *mot1.1 mot2.1* double KO (dKO) and a *mot1.1 mot1.2 mot2.1* triple KO (tKO) showed only slight alterations when grown hydroponically under molybdate deprivation in fresh weight, leaf area production and root length (Fig. S5a–c) compared to molybdate availability. Under the same conditions the *mot1.1* showed a significant reduction of leaf area production as observed recently[7]. Interestingly, the a *mot1.1 mot1.2 mot2.1*

tKO showed a plant survival that was significantly reduced by 45 % compared to control conditions (Fig. S5d). The observed significant reduction in NR activity in the *mot1.1 mot2.1* dKO (Fig. S5e) was comparable to that of the *mot1.1* KO under molybdate deprivation as already described earlier[7]. Interestingly, the *mot1.1 mot1.2 mot2.1* tKO showed a drastically reduced NR activity even under molybdate availability conditions that was further decreased significantly under molybdate deprivation.

**Interplay of MOT2-family with the molybdenum-insertase Cnx1**. Cytosolic Moco-biosynthesis proteins undergo tight protein-protein interactions in a multi-protein complex anchored to actin by Cnx1[24]. Additionally, Cnx1 is supplied with molybdate from the vacuole by interaction with MOT1.2[7]. Therefore, a direct supply of Cnx1 with extracellular molybdate by protein-protein interaction with the MOT2-family was investigated by bimolecular fluorescence complementation (BiFC).

Here, only a C-terminal reporter fusion to MOT2-family members was taken into consideration as topology studies revealed that only the C-terminus is oriented into the cytosol. Aquaporin *At*PIP2A is PM-localized[20] and served as negative control. MOT2.1-VYNE co-expressed with Cnx1-SCYCE showed brighter fluorescence intensity (Fig. 5a1) than the negative control (Fig. 5a2). Abundance controls (Fig. S6) displayed comparable fluorescence intensities indicating that MOT2.1 and *At*PIP2A were present in equal amounts. Results demonstrate interaction of MOT2.1 with Cnx1. Analysis of MOT2.1 and Cnx1 by split-luciferase (split-LUC) experiments underline this interaction and designate the G-domain of Cnx1 as main interaction partner (Fig. S7). BiFC interaction approaches of MOT2.2A (Fig. 5b1) and MOT2.3 (Fig. 5d1) showed higher fluorescence intensities compared to the respective negative controls demonstrating interactions with Cnx1 (Fig. 5b2, d2). Solely the shorter MOT2.2B showed no interaction with Cnx1 (compare Fig. 5c1, c2).

**Molybdate undergoes complex formation with GSH**. Until now it remained unclear how molybdate is imported into the vacuole. Since no member of the MOT2-family was localized in the tonoplast an independent mechanism is suggested. Therefore, we hypothesized complex formation of glutathione (GSH) and the heavy metal (HM) ion molybdate. In general, HMs (e.g. cadmium) are nonspecifically sequestered by GSH, channeled into the vacuole and released[25].

Complex formation of GSH with cadmium ions induces charge transfer of sulfhydryl groups showing absorption in the far ultra-violet (UV) spectral region[26]. To study if such an effect could account for molybdate sequestration, equimolar concentrations of GSH and molybdate were co-incubated and analyzed by spectroscopy (Fig. 6a). After subtraction of the molybdate background, a bathochrome shift from $\lambda_{max} = 221$ nm to $\lambda_{max} = 261$ nm and an increased peak intensity were observed when both components were mixed giving a first indication of GSH-molybdate complexes.

The presence of GSH-molybdate complexes was verified by high-resolution mass spectrometry (HRMS). Mixed solutions of GSH and molybdate show an ion that is not present in pure solutions of both components. The new ion

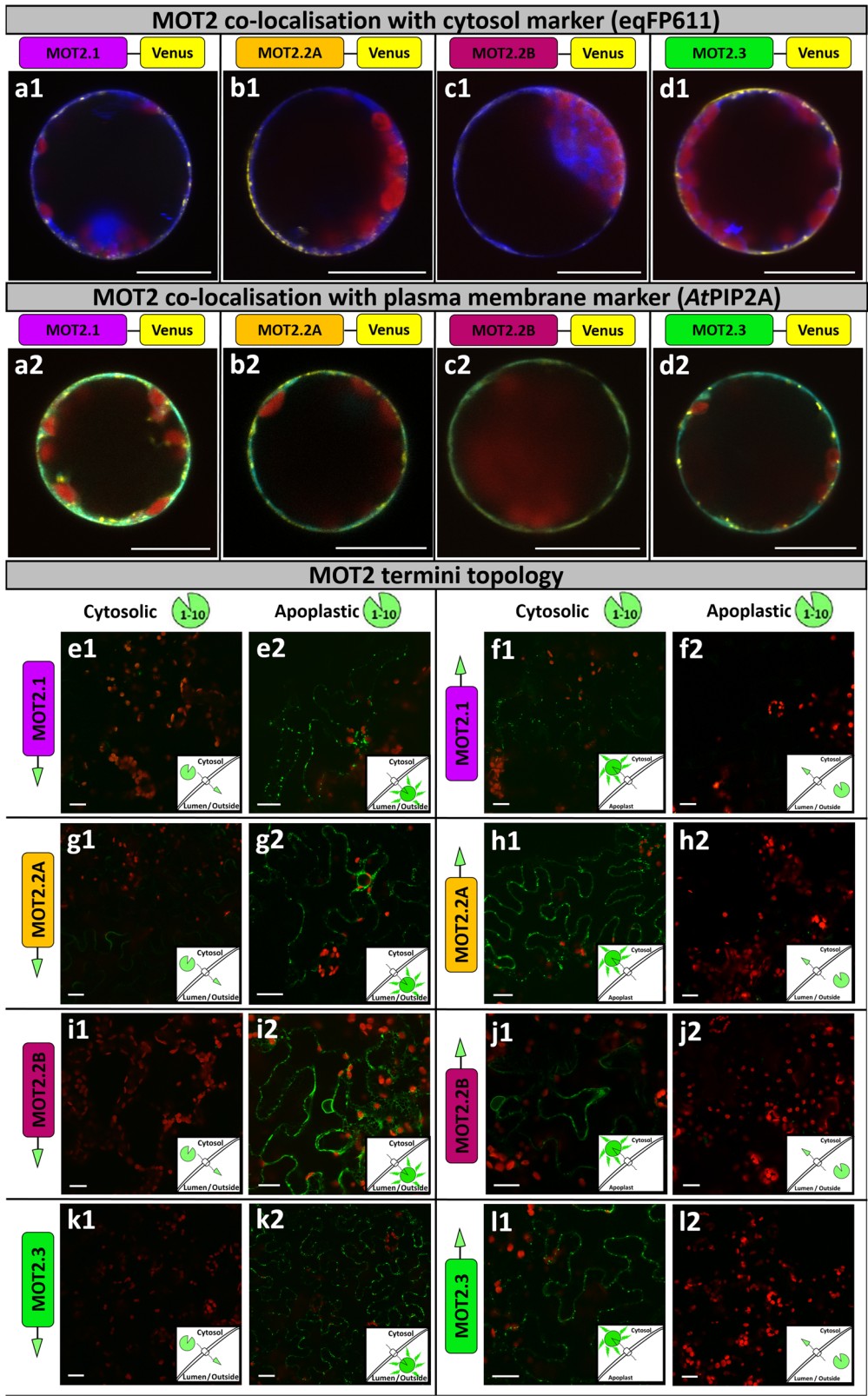

([M-H$^+$] = 433.9561) was assigned the molecular formula C$_{10}$H$_{14}$MoN$_3$O$_8$S$^-$ (Fig. 6b), indicating the formation of a chelate complex with bidentate GSH and molybdate under the elimination of water. This complex formed reproducibly under different molar ratios of GSH and molybdate (1:1 to 9:1) and various pH values (pH = 3–9). Observed isotopic pattern is in agreement with calculated isotopic ratio for a GSH-molybdate complex

(Fig. 6b). The ambidentate character of GSH allows several types of molybdate complexation. Thus, the corresponding ion (m/z = 433.9561) was analyzed by tandem mass spectrometry (MS$^2$) at different collision energies to decipher its structure (Fig. 6c). Use of isotopically enriched ($^{98}$MoO$_4$$^{2-}$) simplified the MS$^2$ spectra, which showed the loss of a fragment consistent with C$_5$H$_7$NO$_3$. This fragment was assigned as glutamine (Fig. 6d),

**Fig. 2 Intracellular localization of MOT2-family members and terminus topology. a–d** Transient chemical transformation of N. benthamiana mesophyll protoplasts with MOT2-Venus fusion constructs and co-expression of eqFP611 as cytosolic marker (a1–d1) or AtTPIP2A-RFP as PM-marker (a2–d2). Images are merge of Venus (yellow), chloroplast auto-fluorescence (red) and marker (eqFP611 in blue, a1-d1; AtPIP2A-RFP in cyan, a2-d2) detection channels. A splitted image, showing each fluorescence channel individually can be found in Fig. S1. **e–l** Split-10 + 1 GFP topology studies by Agrobacterium-mediated transient transformation of N. benthamiana leaves with GFP11-MOT2 (**e, g, i** and **k**) and MOT2-GFP11 (**f, h, j** and **l**). Co-transformation after 2 days with cytosolic GFP-10 (e1-l1) or apoplastic SP-GFP1-10 (e2-l2). Images are merge of GFP (green) and chloroplast auto-fluorescence (red) detection channels. Images were taken after 2-3 days using a C-Apochromat 40x/1.2 water immersion objective (**a–d**) or a Plan-Neofluar 10x/0.3 objective (**e–l**). Scale bars depict 20 μm.

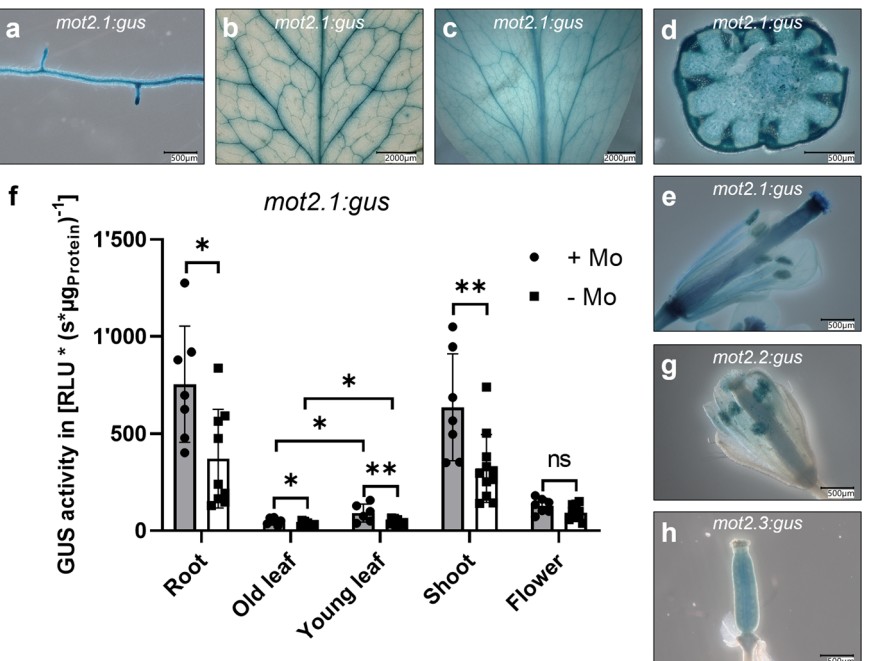

**Fig. 3 Organ-specific expression of the *mot2* genes. a–e** Histochemical GUS assay of *mot2.1:gus Arabidopsis* plants. Depicted are root (**a**), old leaf (**b**), young leaf (**c**), shoot cross-section (**d**) and ovary (**e**). **f** Fluorimetric GUS assay of *mot2.1.gus* plants. Plotted is the mean of 7-10 plants of three independent lines for both conditions. Each organ was analyzed as technical triplicate. Error bars depict standard deviation. Unpaired *t*-test was used to test for significance. ns = not significant, *$P \leq 0.05$, **$P \leq 0.01$. **g** Histochemical assay of *mot2.2:gus* flower. **h** Histochemical assay of *mot2.3:gus* flower. WT shows no background activity (Fig. S3). Images 3**a, b, c, e, g** and **h** are identical to the respective images in Supplementary Fig. S3. Scale bars depict 500–2000 μm as indicated for each panel. Plants for histochemical assay were grown hydroponically under +Mo conditions and for fluorimetric assays under +Mo and -Mo conditions. Quantitative data in **f** was additionally analyzed by two-way ANOVA and Sidak post hoc test for multiple comparison (Table S3).

**Table 1 A. thaliana T-DNA insertion lines.**

| Gene | AGI code | T-DNA insertion line | Source |
|---|---|---|---|
| *mot2.1* | AT4G27720 | GK-086B07.20 | NASC |
| *mot2.2* | AT1G64650 | SALK_054431c | NASC |
| *mot2.3* | AT3G49310 | SALK_077357 | NASC |
| *mot1.1* | AT2G25680 | SALK_118311 | Minner-Meinen et al.[7] |
| *mot1.2* | AT1G80310 | SALK_015044C | Minner-Meinen et al.[7] |
| *mot1.1 mot2.1* double KO | - | - | manual crossing of single KOs |
| *mot1.1 mot1.2 mot2.1* triple KO | - | - | manual crossing of single KOs |

All lines were ordered from the NASC.

indicating that molybdate is complexed by glycine and cysteine. Accordingly, a fragment corresponding to $HMoO_3S^-$ supported the complexation of cysteine by sulfur (Fig. 6c, d). These HRMS experiments suggest that GSH acts as bidentate ligand, chelating molybdate via the carboxylate of glycine and the thiol in cysteine.

## Discussion

Molybdate is a crucial micronutrient for plants and acts as active component of Moco, on which activity of important Moco-enzymes is relying. Molybdate uptake is maintained by specialized MOT proteins. Whereas the physiology of the MOT1-family in *Arabidopsis* is well understood[7], global molybdate homeostasis remained unclear until now. The presence of a MOT2-family in *A. thaliana* was repeatedly postulated, however, their MOT function has never been experimentally proven so far. By in silico studies, three *mot2*-genes encoding four MOT2-proteins with high sequence similarity to the characterized *Cr*MOT2 were identified[8,9]. All of the encoded proteins share

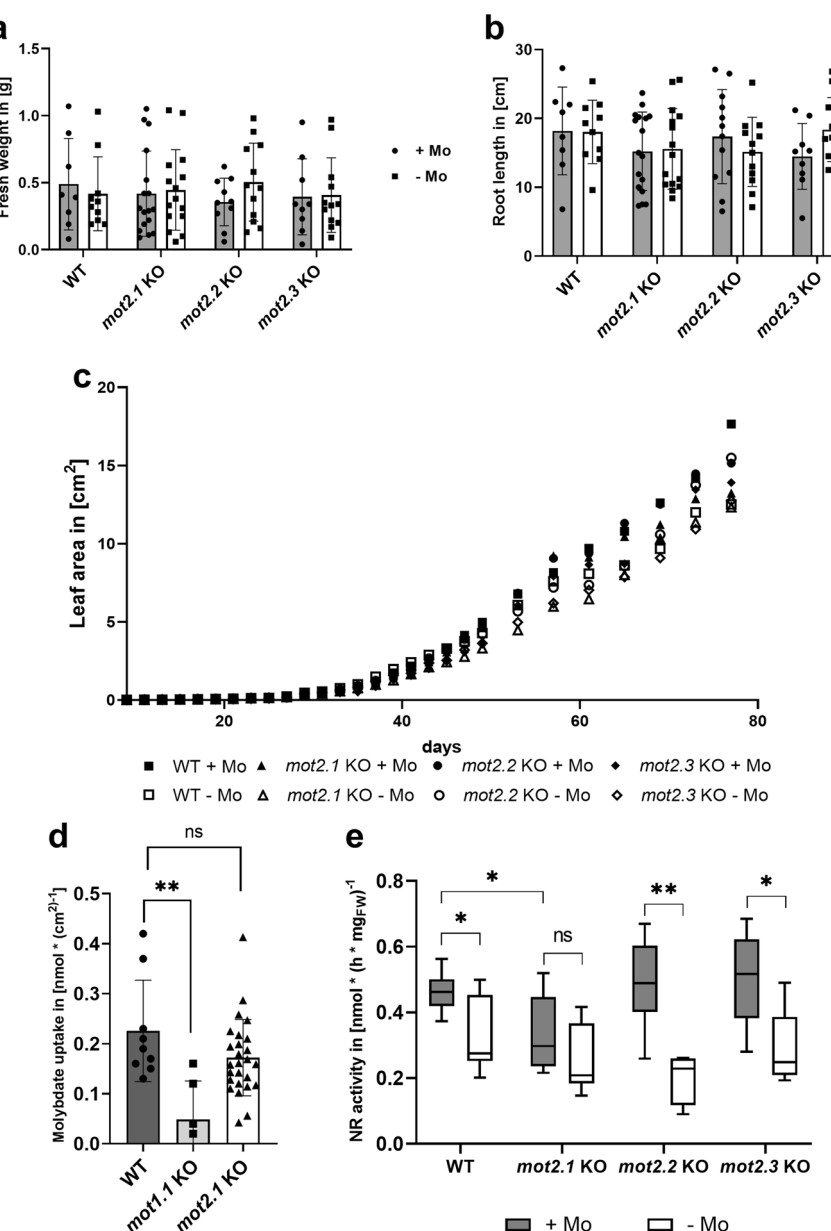

**Fig. 4 Impact of molybdate deprivation on the macroscopic and molecular phenotype of *mot2-KO*.** *Arabidopsis mot2*-KOs were grown under molybdate availability (+/+Mo) and deprivation (−/−Mo) in a hydroponic system for 85 days. **a** Fresh weight of rosette leaves. **b** Root length. **c** Leaf area growth curve within 85 days. Plotted is the mean of 8–15 individuals. **d** Molybdate uptake rate normalized to leaf area of WT, *mot1.1*-KO and *mot2.1*-KO of 56 days old plants. Plotted is the mean of 6–9 individuals for controls and 26 individuals of the *mot2.1*-KO. Quantitative Data was additionally analyzed by one-way ANOVA and Dunnet post hoc test for multiple comparison (Table S4). **e** NR-activity of *mot2*-KOs grown hydroponically for 60 days. Plotted is the mean of 8–15 individuals. Error bars depict standard deviation. Unpaired *T*-test was used for significance tests. ns = not significant, *$P \leq 0.05$, **$P \leq 0.01$. Quantitative data was additionally analyzed by two-way ANOVA and Tukey post hoc test for multiple comparison (Table S5).

four highly conserved motifs essential for molybdate transport[8]. All four MOT candidates imported molybdate into yeast cells with lower capacity than MOT1.1 which has been characterized as high-affinity importer[10]. Furthermore, all MOT2s are localized in the PM. Therefore, MOT2-family members generally maintain molybdate import from extracellular space into cytosol. However, their individual organ-specific expression pattern suggests versatile physiological functions, complementing the MOT1-family.

Recently it was shown that loss of the main radicular molybdate importer MOT1.1 led to dwarfism and dramatically reduced NR-activity under molybdate shortage caused by a reduced molybdate uptake[7]. Observed expression of *mot2.1* in roots could indicate a supportive role in molybdate uptake. However, loss of

MOT2.1 has no impact on molybdate uptake from soil, which points to a different function after molybdate has been taken up by MOT1.1. MOT2.1 might rather be responsible for molybdate distribution within the plant as well as delivery of molybdate to Moco-biosynthesis. This model is supported by a strong expression of *mot2.1* in young leaf tissue, shoot vascular tissue and main leaf venation that is induced by presence of molybdate, by its PM-localization, and by its direct protein interaction with molybdenum-insertase Cnx1. Loss of MOT2.1 affected efficiency of Moco-biosynthesis due to reduced supply with molybdate, observed by reduction of NR-activity. In contrast, lack of MOT2.2 A, MOT2.2B, and MOT2.3 had neither an effect on vegetative growth nor on NR-activity. This leads to the

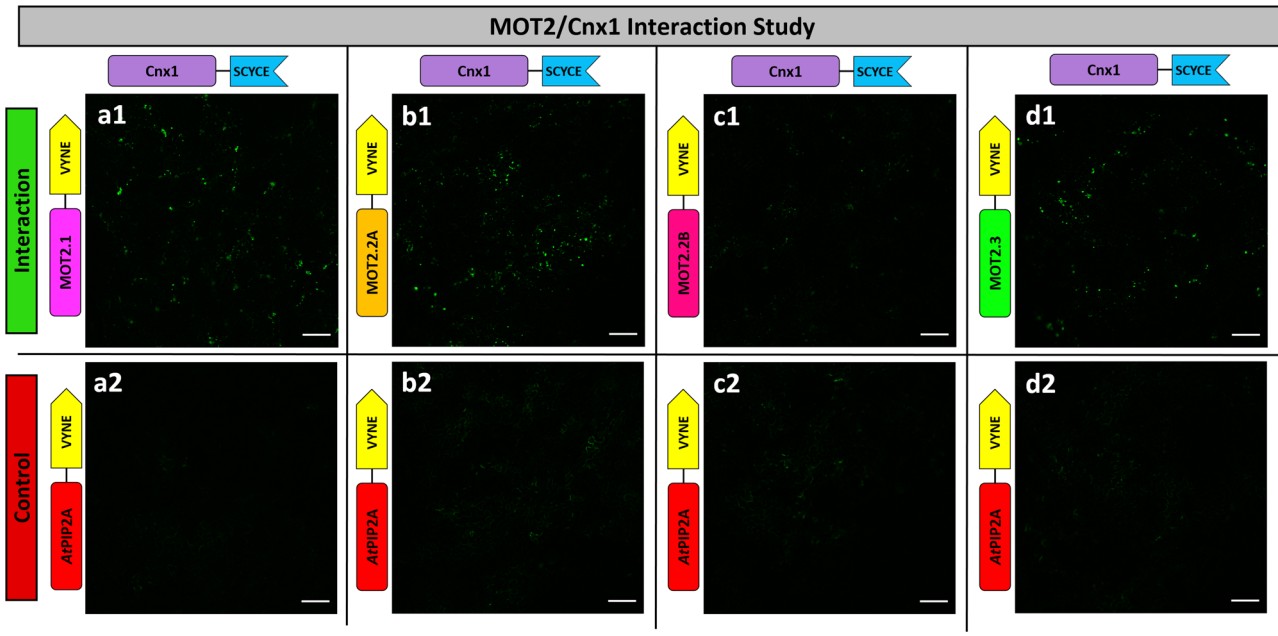

**Fig. 5 Interaction studies of MOT2-family members and Cnx1 via BiFC. a1–d1** Interaction approaches with transiently transformed leaves of *N. benthamiana* co-expressing MOT2-VYNE and Cnx1-SCYCE. **a2–d2** Negative controls with *At*PIP2A-VYNE. Images depict the lower leaf epidermis cell layer and were taken with a Plan-Neofluar 10x/0.3. Scale bars depict 20 μm. Abundance controls, where CLuc replaces Cnx1, showed comparable fluorescence intensities of the proteins of interest and the negative control to display artefacts derived from random interaction (Fig. S6).

conclusion that, even though, an interaction of MOT2.2 A and MOT2.3 with Cnx1 was detected, they only play minor roles in Moco-biosynthesis supply.

Strikingly, all members of the MOT2-family are expressed in the flower. Both, *mot2.1* and *mot2.3*, are highly expressed in ovaries where import of essential nutrients into developing seeds takes place. Taken their organ-specific expression into account, one can assume an involvement in loading molybdate into developing seeds.

In addition of making molybdate available for the daughter generation, high amounts of molybdate in developing seeds might fulfil two additional roles. First, the Moco-enzyme AAO catalyzes the last step of ABA biosynthesis which is the key-hormone in promoting seed dormancy[5]. Although the majority of ABA originates from vegetative tissue and is transported to the seed, ABA is also produced in seed coating tissues[27]. This demand of molybdate for functional Moco can be met by MOT activity. Secondly, high molybdate concentrations (>250 μM) in reproductive organs might be necessary with regard to their inhibitory effect on purple acid phosphatases (PAP)[28]. The main storage form of phosphor in seeds and pollen grains is phytic acid. One member of PAPs, *At*PAP15, is localized in pollen grains and is thought to be the key-phytase during pollen germination to mobilize phosphorous reserves[28]. A genome wide analysis of *pap* genes in *Brassica rapa* revealed potential functions of several PAPs in pollen maturation, germination, and pollen tube elongation[29]. Therefore, it can be assumed that molybdate acts as key-inhibitor on PAPs to avoid premature pollen maturation and germination. Expression of *mot2.2* in pollen grains suggests involvement in this important process.

Elucidation of the physiological roles of the MOT2-family gives a broad insight into molybdate homeostasis. The radicular importer MOT1.1 absorbs molybdate from soil with a strikingly high-affinity[7,10]. In younger plants, MOT2.1 is involved in plant wide distribution, cellular import, and supply of Moco-biosynthesis with molybdate via direct interaction with Cnx1. Previous studies revealed that the vacuole serves as main

storage[12] from where molybdate is exported by MOT1.2 and handed over to Cnx1 for Moco-biosynthesis[7]. A perturbation of this interplay, interestingly, has a severe effect on vegetative growth. The simultaneous loss of the main radicular importer MOT1.1, the main cellular importer MOT2.1, and the tonoplast exporter MOT1.2 showed a drastic reduction in NR activity culminating in a strongly reduced plant survival rate independent of molybdate presence.

Since free molybdate as HM-ion has a hazardous potential[25], the route of molybdate throughout the plant raises an interesting question: How is molybdate imported into the vacuole after cellular import? One general coping mechanism is the conjugation of HM-ions with GSH mediated by GSH-S-Transferases[30]. The conjugate is imported to the vacuole and dissociates due to pH shifts[25]. Therefore, we hypothesize that excess molybdate is sequestered by GSH to avoid toxic processes. This model was supported by a bathochrome shift in absorption caused by formation of GSH/molybdate-complexes as shown for several HM/GSH-complexes[26,31] and additionally evidenced by mass spectrometry experiments. These results support a model in which molybdate is channelled in conjugation with GSH into the vacuole by unspecific HM/GSH transporters[32]. There, the complex dissociated due to the pH shift so that molybdate can be stored and exported by MOT1.2 when needed for Moco-biosynthesis. All MOT2-family members are expressed in the reproductive organs to import molybdate. Both splice variants of MOT2.2, thereby, might fulfil different roles: While MOT2.2 A directly interacts with Moco-biosynthesis to supply AAO, MOT2.2B is not interacting with Moco-biosynthesis and might supply free molybdate for PAP inhibition and seed nutrition storage.

Recently, MOT2-family proteins were implicated as SAM-transporters located in the Golgi[14]. In this study, the double KO of *mot2.1* (*GoSAMT2*) and *mot2.2* (*GoSAMT1*) showed reduced Golgi synthesized polysaccharide methylation and changed cell wall architecture. Furthermore, fluorescence signals of MOT2-GFP co-localized with the GFP-tagged Golgi marker *Gm*Man-1.

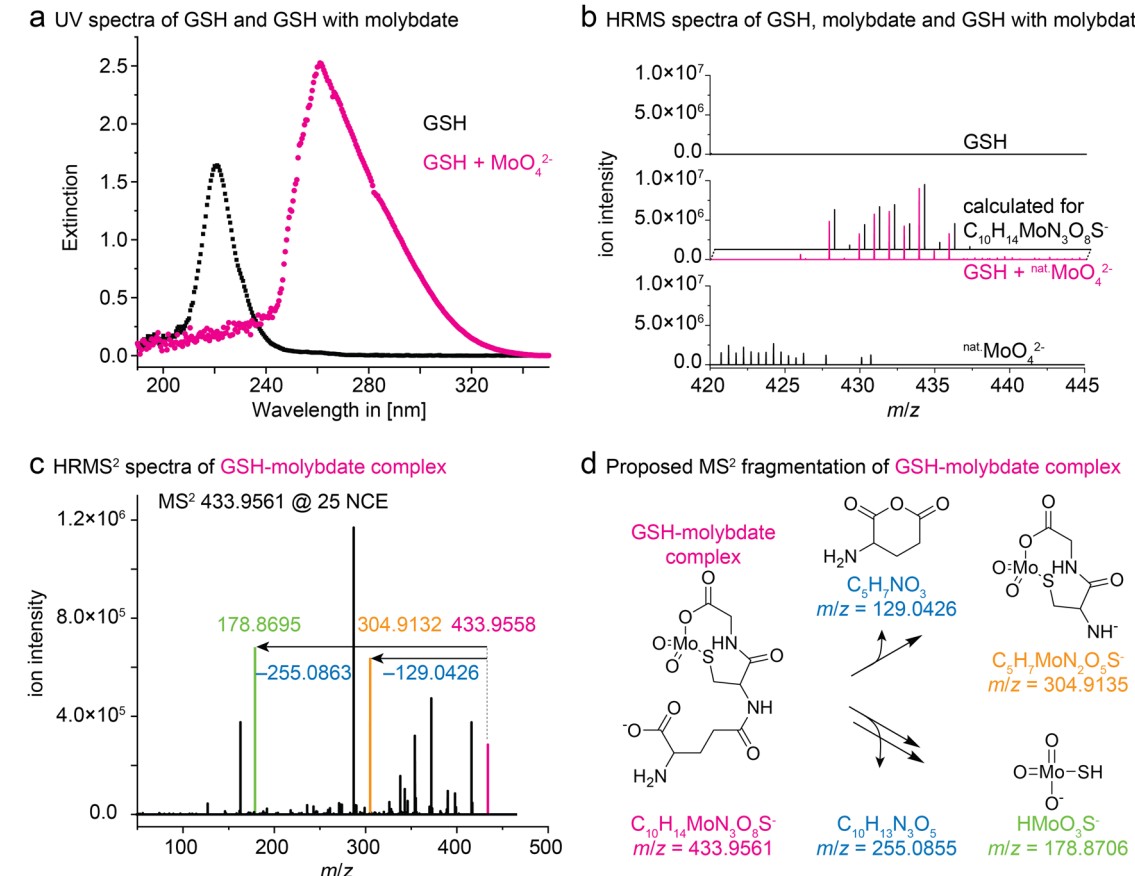

**Fig. 6 Characterization of a GSH-molybdate complex by UV spectroscopy and HRMS. a** UV spectra of GSH and GSH with molybdate. **b** HRMS spectra of GSH, molybdate and a GSH-molybdate mixture recorded in negative mode. Shown are absolute ion intensities in the range of $m/z = 420$–445. No dominant ions are present in the GSH sample. Unassigned ions in the molybdate MS possibly originate from poly-molybdates. The solution containing both molybdate and GSH shows new ions that are not present in the stock solutions containing solely GSH or molybdate. Comparison with calculated isotopic pattern of a GSH-molybdate complex sum formula $C_{10}H_{14}MoN_3O_8S^-$. **c** MS$^2$ fragmentation of $C_{10}H_{14}MoN_3O_8S^-$ with 25 normalized collision energy. Experimentally observed fragment masses were in agreement with calculated ones (Panel d). **d** Proposed fragmentation of a molybdate chelate complex with bidentate GSH, matching the ions observed in MS$^2$ experiments (Panel c). Listed are exact masses and respective sum formulas.

Results of the present study confirmed Golgi localization of all MOT2-family proteins but also showed their clear PM-localization and, most importantly, molybdate transport activity. Given their different localization and different substrates one can assume moonlighting character for the MOT2-family by fulfilling more than only one function[33]. Activity of MOT2-family proteins in Golgi and PM is likely, since the Golgi is a station of MOT2-family members on their way to their final destination, the PM[34]. Also for other members of Moco-metabolism moonlighting activity was demonstrated[35]. For example, gephyrin, the human homologue of Cnx1, is involved neuronal transporter/microtubule interaction[36], while Moco binding proteins are also involved in cytokinin-biosynthesis[37]. Therefore, the moonlighting transporters of the MOT2-family are of pivotal importance for SAM-import into Golgi and plant wide molybdate homeostasis

## Methods

**Moco-yeast strains**. Molybdate transport activities of MOT2 candidates were tested in yeast. As baker's yeast *Saccharomyces cerevisiae* has lost molybdenum metabolism during evolution and hence harbors no Moco-enzymes[16] it is a suitable model organism to test for molybdate transport, as described in detail in Table S1.

**Plant material**. *N. benthamiana* WT plants were used for localization, topology and BiFC analysis. WT *A. thaliana* Col-0 ecotype was used to generate stable transgenic lines carrying endogenous *mot2:gfp-gus* constructs and as control plants. *A. thaliana* T-DNA insertion KO lines (Table 1) were maintained from NASC (Nottingham, UK). Presence of T-DNA insertion at correct loci and homozygotic

status of used KO lines were verified with genotyping PCRs. Higher degree KOs were generated by manually crossing of single KO lines. The status of molybdate availability between lines purchased from NASC and of that generated by manually crossing could not be validated.

**In silico analysis**. *Atmot2.1* was identified according to Tejada-Jiménez[8] using TAIR[38]. Analysis of *mot2.1* with EnsmblPlants[39] revealed two highly-related paralogues in *A. thaliana* (AT1G64650 and AT3G49310). A ClustalW alignment of the MOT2-family using MEGA11[40] was performed to analyze sequence similarity. Alignment was visualized using BioEdit software[41]. The sequence of *mot2.2b* was excluded from this analysis due to full alignment with *mot2.2a*.

**In vivo yeast based growth assay**. Molybdate transport activity of MOT2 candidates was investigated by an in vivo growth assay using transgenic Moco-yeast strains (Table S1). Growth medium pH 5.8 consisted of yeast-nitrogen-base medium (6.7 g/L; Carl Roth, Karlsruhe, Germany) was supplemented with 10x dropout solution (200 mg/L arginine, 300 mg/L isoleucine, 300 mg/L lysine, 200 mg/L methionine, 500 mg/L phenylalanine, 2000 mg/L threonine, 300 mg/L tyrosine, 1500 mg/L valine, 100 mg/L L-adenine-hemisulfate, 200 mg/L histidine, 1000 mg/L leucine, 400 mg/L tryptophan) diluted to 1x concentration and 0.2% (w/v) fructose (Duchefa Biochemie, Haarlem, Netherlands). After shaking at 200 rpm for 48 h at 30 °C, 10 mL of pre-culture inoculated with a Moco-yeast strain was added to 190 mL growth medium for main-culture and incubated analogously. Subsequently, yeast suspension was filled into 50 mL tubes and centrifuged for 15 min at 5200 × g. Supernatant was dismissed and the pellet was split into 1 mL reaction tubes, followed by centrifugation for 5 min at 5200 × g. After dismission of the supernatant, the pellet was frozen in liquid nitrogen. Galactose (Duchefa Biochemie, Haarlem, Netherlands) was added to a final concentration of 2% (w/v) to growth medium containing 250 mM sodium chlorate (NaClO$_3$; Carl Roth, Karlsruhe, Germany) and 100 nM sodium molybdate (Na$_2$MoO$_4$; Sigma Aldrich,

St. Louis, MI, USA) for induction of *mot* gene expression. For non-induction conditions the same volume of water was added instead. Growth medium was inoculated to an $OD_{600}$ of 0.1 with transgenic Moco-yeast strains. A control approach contained no sodium chlorate. A 48-well MTP flower plate (m2p labs Beckman Coulter, Aachen, Germany) was loaded with 1 mL of cell suspension per well in duplicates and sealed with semi-permeable foil (F-GPR48-10, m2p labs Beckman Coulter, Aachen, Germany) to allow gas exchange. The flower plate was loaded to a BioLector® I (m2p labs Beckman Coulter, Aachen, Germany) allowing incubation of the plate shaking at 1200 rpm for 7 days at 30 °C and while simultaneous monitoring of biomass increase. Therefore, light scattering at 620 nm was detected as relative light unit (RLU)-output as biomass index. A biomass threshold of 40 RLU was selected to allow comparison of different strains and to further analyze the growth behavior. In control approaches it was shown that the influence of single growth assay components (sodium chlorate, sodium molybdate and sugar combinations) on the growth behavior of each Moco-yeast strain was neglectable.

**Molecular cloning and transformation for localization, topology, BiFC and Split-Luc studies**. The coding sequences (CDS) of *A. thaliana* genes *mot2.1* (AT4G27720), *mot2.2a* (AT1G64650.1), *mot2.2b* (AT1G64650.2) and *mot2.3* (AT3G49310) were amplified by Phusion polymerase (Thermo Scientific, Waltham, MA, USA) from cDNA of *A. thaliana* with *attB* site flanked primers to use the GATEWAY cloning system (Invitrogen, Waltham, MA, USA). PCR fragments were inserted into the pDONR/Zeo vector by BP reaction generating pEntry-*mot2* entry vectors.

For localization studies the resulting entry vectors were recombined with pDest-*GW-venus* vectors using LR reaction to generate expression vectors coding for *mot2-venus* fusion constructs. As cytosolic marker eqFP611[42] was used. To create a PM-marker, the CDS of *Atpip2a* (AT3G53420) from *A. thaliana* was amplified with *attB* site flanked primers and sub cloned by BP reaction into the pDONR/Zeo vector. Recombination with pDest-*GW-rfp*[43] by LR reaction led to the expression vector pExp-*Atpip2a-rfp*. The Golgi marker *man1* from soybean (*Glycine max*) was synthesized from two oligonucleotides that were annealed and amplified in a first PCR. A second PCR with *attB* site flanked primers allowed to use the fragment in a BP reaction to create an entry vector. A LR reaction with pDest-*GW-rfp* generated the expression vector pExp-*GmMan-1-RFP*.

Topology studies of the MOT2 family were carried out using the GATEWAY-based Split-10 + 1 GFP system[22]. Vectors were kindly provided by Prof. Thordal-Christensen from University of Copenhagen. LR reaction of pEntry-*mot2* vectors and pDest-*GW-gfp11* and pDest-*gfp-GW* expression vectors were generated coding for constructs with GFP11 either fused to the N- or the C-terminus of MOT2. The GFP1-10 fragment was co-expressed with either cytosolic localization (GFP1-10) or with localization in the apoplast by using the signal peptide of *At*WAK2 (*A. thaliana* Wall Associated-Kinase 2, AT1G21270) resulting in SP-GFP1-10[44].

For BiFC assays, entry vectors with CDS of *mot2* and *Atpip2a* were recombined by LR reaction with the pDest-*GW-vyne* destination vector resulting in expression vectors coding for *mot2-vyne* and *Atpip2a-vyne* constructs. The C-terminal half of the cytosolic Luciferase (CLuc)[24] was used as abundance control. The CDS was amplified by Phusion PCR with *attB* site flanked primers and used in a BP reaction to generate entry vectors. Those were used in LR reaction with pDest-*GW-vyne* and pDest-*GW-scyce* to generate pExp-*Atpip2a-vyne* and pExp-*nsp3-scyce* expression vectors. Generation of pExp-*cnx1-scyce* expression vectors was generated by LR reaction with the pDest-*GW-scyce* destination vector and the entry vector pEntry-*cnx1* (AT5G20990)[43].

For split-Luc analyses LR reaction using pEntry-*mot2.1* and pDest-*GW-cluc* generated pExp-*mot2.1-cluc* expression vectors. Expression vectors pExp-*cnx1-nluc*, pExp-*cnx1e-nluc*, pExp-*cnx1g-nluc*, negative controls pExp-*nsp3-nluc* and abundance control pExp-*scyce-nluc* were generated by cloning via LR reaction the entry vectors pEntry-*cnx1* (AT5G20990), pEntry-*nsp3* (AT3G16390) and the BiFC-terminus pEntry-*scyce* into the destination vector pDest-*GW-nluc*[45]. Expression vector pExp-*Atpip2a-cluc* was generated by LR reaction with pEntry-*Atpip2a* and pDest-*GW-cluc*. Primers used to generate the vectors (Table S6) of this study are listed in Table S7. Plant transformation for localization, topology, BiFC and split-Luc assay was carried out according to Minner-Meinen et al.[7]. Expression in *N. benthamiana* leaves was carried out by *Agrobacterium tumefaciens* mediated transformation[45–47]. Protoplast preparation and chemical transformation were carried out according to Negrutiu et al.[48]. *A. thaliana* seedlings were transformed by the FAST (fast agro-mediated seedling transformation) method[49].

**Cultivation of plants in soil**. Germination of *N. benthamiana* was achieved in common potting soil at 22–25 °C with 10 h of artificial light (≈ 60 µE) per day under greenhouse conditions and sufficient water supply. Seedlings were transferred two weeks after germination, into 9 × 9 cm pots containing pot soil mixed with 1% NPK-fertilizer (Blaukorn® classic, Compo Expert, Münster, Germany) and 5% Perlite. Five to 12 weeks old plants were used for experiments. *A. thaliana* seeds were stratified at 4 °C for 48–72 h. Two weeks after germination on common pot soil, seedlings were transferred into 5 × 5 cm pots and cultivated at 22–25 °C in a walk-in phyto-chamber with 10 h of artificial light (≈ 60 µE) per day and 60–70% humidity.

**Hydroponic growth system**. Plants were grown in a hydroponic growth system[50] modified according to Minner-Meinen et al.[7]. Germination and basal nutrient solution (¼ Hoagland´s solution) with a pH value of 5.6 were prepared either with 100 nM of sodium molybdate ( + Mo, molybdate availability) or by excluding sodium molybdate from the original recipe (-Mo, molybdate deprivation). *A. thaliana* seeds were sown on microcentrifuge tube lids with a pin-hole and placed in a germination tank. After 20 days plants were transferred to aerated tanks and harvested after a total time span of 60 days. From germination tanks seedlings were also transferred to 130 mL centrifuge tubes and grown under molybdate availability conditions for the molybdate uptake assay.

**Generation of multi-KO plant lines**. Petals and stamen of a flower of the mother plant carrying one T-DNA insertion KO were removed to avoid the self-fertilization of the ovary. Stamen harboring mature pollen from a father plant carrying a second T-DNA insertion KO was used to fertilize the prepared ovary of the mother plant. The fertilized ovary was brought to seed maturation. The harvested seeds were grown under antibiotics selection and the presence of the inherited T-DNA insertion was tested using genotyping PCR.

**Generation of stably transformed *mot2:gus* A. thaliana lines**. Endogenous promoters were defined as the region of 1998 bp (*mot2.1*), 1924 bp (*mot2.2*) and 1980 bp (*mot2.3*) upstream of each start codon. Regions were amplified from genomic DNA of *A. thaliana* by phusion PCR with *attB* site flanked primers. Fragments were sub-cloned using the GATEWAY cloning system (Invitrogen, Waltham, MA, USA) into pDONR/Zeo via BP reaction generating entry vectors. By recombination via LR reaction with pKGWFS7[51] expression vectors coding for a GFP-GUS-fusion construct expressed under the control of the endogenous *mot2* promoters (*mot2.1:gus*, *mot2.2:gus* and *mot2.3:gus*) were generated. To generate stable *A. thaliana* lines, floral dip[52] with *Agrobacterium tumefaciens* was carried out. Selection for transformants in T0 and T1 generation was carried out with kanamycin (Duchefa Biochemie, Haarlem, Netherlands). To avoid positional effects of random T-DNA integration after *Agrobacterium* transformation, three individually generated lines were used for further experiments. The presence of the *mot2.gus* constructs was tested by PCR using genomic DNA.

**Histochemical and fluorimetric GUS assay**. Histochemical and fluorimetric GUS assays are well suited to gather both, qualitative as well as quantitative information about organ-specific gene expression of the *mot2*-family members. Assays were carried out according to Minner-Meinen et al.[7]. Transgenic *A. thaliana* *mot2:gus* lines were grown hydroponically under molybdate availability for histochemical staining. Plant material was incubated with GUS staining solution[53] via vacuum chamber and incubated at 37 °C over night. After chlorophyll extraction with 70% (v/v) ethanol, documentation was carried out using a Keyence VHX digital microscope (Keyence, Osaka, Japan). WT plants used as control showed no background signal in the analyzed organs (Fig. S3).

Fluorimetric assay was performed with *mot2.1:gus* plants grown hydroponically under molybdate availability and deprivation. Plant material was separated into roots, young leaves, old leaves, shoot and flower during harvest and protein extraction was carried out in triplicates. Samples were loaded into a 96-well plate and mixed with GUS reaction solution containing methylumbelliferylglucuron (MUG; Duchefa Biochemie, Haarlem, Netherlands) that can be cleaved by GUS into the fluorescent product MU (methylumbelliferyl; excitation: 365 nm, emission: 455 nm). Fluorescence was measured using a Tristar LB941 multimode reader (Berthold Technologies, Bad Wildbad, Germany) over a time span of 40 min. GUS activity was calculated as the rate of gain in fluorescence intensity over time normalized to the amount of total protein measured via Bradford assay using Roti®Quant reagent (Carl Roth, Karlsruhe, Germany). WT plants were used as negative control and showed no fluorescence signal in the analyzed organs.

**Phenotype analysis during growth and at harvest**. The leaf area of *A. thaliana* plants growing hydroponically was measured using Easy Leaf Area software[54]. A red square with an area of 4 cm2 was used as reference. By setting the number of green pixels into relation to the known number of red pixels the leaf area was determined in a non-invasive and rapid manner. For 60 days pictures were taken as triplicates with a Panasonic Lumix DMC-GX80 and a Panasonic H-FS 1442 A camera lens (Panasonic, Kadoma, Japan) every 2 days. Furthermore, the developmental stages of the plants were recorded[55] every 2 days and overall plant survival was analyzed. After 60 days plants were harvested and fresh weight as well as root length were measured.

**Molybdate uptake assay**. *A. thaliana* (8 weeks old) were grown in 130 mL tubes filled with molybdate containing basic nutrient solution. After two weeks the medium was harvested in 50 mL duplicates and leaf area was determined. Molybdate concentrations were measured in a modified manner according to Cardenas and Mortensen[56]. A sample volume of 50 mL lowered the detection limit of the assay to 10 nM molybdate. Samples were mixed thoroughly with 50 µL of sulfuric acid and 250 µL of assay reagent (2% (*w/v*) sodium hydroxide, 2 g/L 1,2-Dimercapto-4-methylbenzene, Thermo Scientific, Waltham, MA, USA; 16 mL/L thioglycolic acid, Supelco, Sigma Aldrich, St. Louis, MI, USA) for 1 min and shaken

for 20 min. After adding 1 mL of pure isoamyl acetate, mixing for 2 min and shaking for another 20 min, the organic phase was extracted from the conical end of the tube. Extinction was measured in a solvent-resistant cuvette at 680 nm blanked against pure solvent. A calibration curve from 0 to 150 nM sodium molybdate enabled in basic nutrient solution was measured to determine molybdate concentration. The amount of absorbed molybdate was calculated and normalized with the leaf area. WT plants and *mot1.1*-KO plants served as positive and negative control, respectively.

**NR enzyme activity assay**. The NR enzyme activity assay[57] was carried out according to Minner-Meinen et al.[7]. Leaf material of *mot2*-KO lines grown hydroponically under molybdate availability and deprivation was harvested and homogenized under liquid nitrogen cooling. Homogenate was mixed with extraction buffer, mixed thoroughly and centrifuged. Assay reaction was started by mixing the extract with assay buffer and stopped after 10 min, 20 min, and 30 min by adding zinc acetate. The amount of formed nitrite was measured colorimetrical after reaction with sulfaniamid (SA; Merck, Darmstadt, Germany) and N-[Naphthyl-(1)]-ethylendiammoniumchlorid (NED; Sigma Aldrich, St. Louis, MI, USA). NR enzyme activity was calculated in nmol $NO_2^-$ per mg of fresh weight and hour.

**UV-Spectroscopy of GSH-molybdate complexes**. Stock solutions of reduced GSH (Duchefa, Harlem, Netherlands) and sodium molybdate (Sigma-Aldrich, St. Louis, MI, USA) were prepared in water with a concentration of 2 mM. The 2 mM GSH sample was diluted with water to a concentration of 1 mM. The spectrum of each sample was recorded by measuring the extinction of light in the UV region ranging from 190 nm to 350 nm in special UV cuvettes with a layer thickness of 1 cm using an Ultrospec 2100 pro UV/VIS spectrophotometer (Amersham Biosciences, Amersham, UK). Subsequently, the background of a sodium molybdate solution with a concentration of 1 mM was subtracted from the spectrum by blanking. The GSH and molybdate solution was mixed equimolar to a concentration of 1 mM each and the extinction spectrum was recorded.

**Liquid chromatography mass spectrometry**. Liquid chromatography mass spectrometry (LC-MS) measurements were performed using Q Exactive Orbitrap (high-performance benchtop quadrupole Orbitrap mass spectrometer; Thermo Fisher Scientific, Bremen, Germany) HRMS with electrospray ion source (mass range in negative mode: $m/z = 100$–1500) and Ultimate3000 UHPLC system (Thermo Fisher Scientific, Bremen, Germany). Samples were either injected by direct injection (flow rate was 0.6 mL·min$^{-1}$) or by UHPLC. The following UHPLC conditions were used: An Accucore C18 column (2.1 × 100 mm, 2.6 μm, Thermo Fisher, Bremen, Germany) was used with gradient elution as follows: MeCN (0.1% ($v/v$) HCOOH)/H$_2$O (0.1% ($v/v$) HCOOH) initially at 5:95, reaching 2:98 over 7 min, then maintaining 2:98 for 3 min. The flow rate was 0.2 mL·min$^{-1}$ and injection volume was 3 μL. Isotopically enriched molybdenum (enrichment >98%) was purchased from Eurisotop (Saint-Aubin Cedex, France). The molybdenum oxide $^{98}$MoO$_3$ was dissolved in sodium hydroxide solution to maintain a sodium molybdate solution. Subsequently, the prepared isotope solution was diluted with deionized water to maintain a stock solution containing $10^{-2}$ mol L$^{-1}$ molybdate.

**Identification and characterization of the GSH-molybdate complex**. Three separate stock solutions (1 mL each) with Na$_2$MoO$_4$, Na$_2$$^{98}$MoO$_4$ and GSH at a concentration of $10^{-2}$ mol·L$^{-1}$ in water were prepared. GSH stock solution (100 μL) was diluted with water (900 μL) prior to the MS measurement. Na$_2$MoO$_4$ stock solution (100 μL, 1 eq.) and GSH stock solution (100 μL, 1 eq.) were mixed and stored at room temperature (~22 °C) for 1 h. Prior to the MS measurement, the solution was diluted with water (1800 μL). The Na$_2$MoO$_4$ stock solution (100 μL) was diluted with water (900 μL) prior to the MS measurement. All samples were measured with HRMS by direct injection. Subsequently, solutions containing both molybdate and GSH were prepared with molar ratios of 1:1 to 1:9. Upon MS measurement, identical results were maintained, i.e. $m/z = 433.9556$ was detected in all samples. In addition, the pH values of the solutions were set to 3, 5, 6, 7, and 9 and measured by HRMS. In all cases, the same ion with $m/z = 433.9556$ was observed.

**Bi-molecular fluorescence complementation assay**. Protein-protein interaction was analyzed using bi-molecular fluorescence complementation assay[43,47] according to Minner-Meinen et al.[7]. To allow comparable results the lower epidermis of 5–10 leaf discs from 2–3 *N. benthamiana* plants was analyzed with identical cLSM settings. The interaction approach included MOT2-VYNE and Cnx1-SCYCE fusion constructs expressed in one leaf half. The negative control consisted of *At*PIP2A-VYNE and Cnx1-SCYCE expressed in the other leaf half. An abundance control was carried out where Cnx1-SCYCE was exchanged with the C-terminal half of cytosolic luciferase (Luc) fused to SCYCE (CLuc-SCYCE) to allow the estimation of different concentration levels of the negative control protein in correlation with the interaction approach counterpart and resulting random fluorescence signals[43].

**Microscopy detection**. Localization, topology and BiFC studies were carried out using a confocal Laser Scanning Microscope (cLSM) LSM 510 Meta from Zeiss (Göttingen, Germany)[7,46]. The cLSM-510META scan head was connected to Axiovert 200 M. All objects were analyzed either with a Plan-Neofluar 10x/0.3 or a C-Apochromat 40x/1.2 water-immersion objective. Excitation was carried out using both an argon laser (488 nm line for all VENUS approaches, as well as chlorophyll auto fluorescence) or a Helium-Neon Laser (543 nm line for eqFP611). Primary beam splitting was achieved by UV/488/543/633 mirrors. Secondary beam splitters acted at 545 nm. Filter sets for the detection of fluorescence was used as followed: BP 505-530 nm for all split-GFP (Em$_{max}$: 510-515 nm) and VENUS (Em$_{max}$: 525 nm) approaches; BP 560-615 for eqFP611 (Em$_{max}$: 611 nm); LP 650 nm for chlorophyll auto fluorescence. The Lambda mode was used to examine spectral signature of all fluorophores. All images were taken using ZEISS Microscope Software ZEN 2009.

**Split-Luc protein-protein interaction assay**. Split-Luc assay to analyze protein-protein interaction was performed by floated leaf luciferase complementation imaging (FLuCI) according to Kaufholdt et al.[45]. Interaction approach consisted of leaves co-expressing MOT2.1-CLuc and either Cnx1-NLuc, Cnx1E-NLuc or Cnx1G-NLuc. Negative control replaced either MOT2.1-CLuc with *At*PIP2A-CLUc or Cnx1-constructs with NSP3-NLuc. Abundance control to depict signals originating from random interaction consisted of SCYCE-NLuc and replaced Cnx1-constructs. Interaction approach and negative control or abundance control were infiltrated into one leaf half each. After 2–7 days of transformation, 6 leaf discs (12 mm diameter) per leaf half from 4 leaves of 4 plants each were infiltrated with luciferin solution (10 mM MES pH 5.6, 10 mM MgCl2, 0.5% ($v/v$) DMSO, 0.1 mM luciferin). Luminescence was measured as RLU using a Tristar LB941 multimode reader (Berthold Technologies, Bad Wildbad, Germany) at 560 nm for 20 min. Interaction approach split-Luc factor was calculated by dividing the mean of the interaction approaches with the negative controls from the other leaf half. Abundance control split-Luc factors was calculated analogously. Interaction takes place, when the interaction approach split-Luc factor is higher than the abundance control split-Luc factor.

**Statistics and reproducibility**. All experiments were performed with multiple independent samples as described in Materials and Methods and the figure legends. Experiments addressing localization, termini topology, BiFC, Split-Luc and GUS-staining were performed multiple times and had similar results to those reported here. Growth kinetics of Saccharomyces and plants are repeated at least three times. Means of independent experiments are plotted and error bars depict standard deviation. As stated in each figure/table legend, unpaired $T$-test was used to test for significance, two-way ANOVA analyses and Sidak post hoc tests for multiple comparison were used to analyze quantitative data. No data were excluded from results reported and all the biological replicates were used for statistical analyses.

**Reporting summary**. Further information on research design is available in the Nature Portfolio Reporting Summary linked to this article.

## Data availability

All data supporting the findings of this study are available within the paper and its Supplementary Information. The numeric data used in the present work can be found in Supplementary Data 1 and are available from the corresponding author on reasonable request.

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

## Acknowledgements

We would like to thank Mattes Hintmann and the work group of Rebekka Biedendieck from the BRICS (TU Braunschweig, Germany) for help with the BioLector® and the supply with flower plates. We would like to appreciate the help of Prof. Hans Thordal-Christensen (University of Copenhagen, Denmark) for providing Split-10 + 1-GFP vectors. *Arabidopsis* T-DNA KO lines were kindly provided by the Nottingham *Arabidopsis* stock center (NASC, Nottingham, UK). Furthermore, we would like to thank Prof. Dr. Erwin Grill for help with the topic of heavy metals and glutathione. We also would like to thank the work group of Prof. Dr. Christian Hertweck for great help with mass spectrometry analyses and for sharing their facility with us. We would like to thank Nele Fiene, Moritz Friesch, Benedikt Gierling, Linda Hage, Melanie Heidecke, Jannik Heiligenstadt, Rena Hinrichs, Saskia Kell, Eike Kreitz, Jan-Hendrik Lenzen, Samuel Meckoni, Lena Meißner, Paul Meyfarth, Merve Saudhof, Fynn Schilling, Ina Schmidt, Alexa Schubert, Nico Sprotte, Luca Steinbacher, Claudia Strauch, Kaja Tünnermann, Hanna Willenbockel, Jess Arnold Siani Wouachi and Chris Zaydowicz for their help, their high commitment and great technical work in our lab. Our special thanks got to our technicians Kristin Eckhoff and Tanja Linke. This work was financially supported by the Deutsche Forschungsgemeinschaft (grant GRK2223/1) to R.H. and R.R.M.

## Author contributions

J.N.W., R.M.M., R.H. and D.K. planned and designed the project. RH and RRM acquired funding. R.M.M., M.B. and J.N.W. performed in silico analysis. TWH, RRM, D.K. and J.N.W. generated yeast strains. SS, J.N.W., D.K., R.H. and R.B. performed yeast growth assays and discussed the results. R.M.M. designed and generated vectors for localization, topology and BiFC. J.S. performed protoplast isolation and transformation. R.M.M., M.B.

and J.N.W. carried out localization studies. R.M.M. performed topology studies. R.M.M. and J.N.W. generated *mot2:gus* plant lines. Lv.d.H. carried out histochemical GUS assays. J.N.W. carried out fluorimetric GUS assays. Lv.d.H. and J.N.W. handled hydroponical growth, phenotype analysis. J.N.W. carried out molybdate uptake assays. Lv.d.H. performed NR assays. R.M.M., L.K. and D.K. carried out BiFC analyses. J.N.W. performed UV-spectra analysis of GSH-molybdate complexes. C.H. and V.H. carried out MS experiments. All authors analyzed and discussed the results. J.N.W. and D.K. were primarily involved in drafting the manuscript. J.N.W., V.H., D.K., Lv.d.H., L.K. and R.R.M. produced figures and tables. J.S., R.H. and R.R.M. critically read the manuscript and improved the text; all authors finalized it. J.N.W., D.K. and R.H. coordinated the work. All authors have read and agreed to the published version of this manuscript.

## Funding

## Competing interests
The authors declare no competing interests.
