## [Peer Review File · Communications Biology]

Reviewers' comments:

Reviewer #1 (Remarks to the Author):

The paper presented a study of the regulatory mechanism of plasma membrane and Golgi-localized MOT2s function in plant molybdate homeostasis. MOT2s was found to interact with the molybdenum-insertase Cnx1 that mediated Moco-biosynthesis proteins complex anchored to actin. This work also revealed GSH was able to bind molybdate via the carboxylate of glycine and the thiol in cysteine to form a complex, which might be responsible for molybdate homeostasis through promoting loading into the vacuole by unspecific HM/GSH transporters. It is concluded that MOT2s is required for molybdate transport in leaves and reproductive organs, thus maintaining molybdate homeostasis in plant tissues. It is therefore important to understand the MOT2 involvement in the regulation of molybdate homeostasis, which plays an important role in SAM-import into Golgi and plant wide molybdate homeostasis.

The paper is concise and overall well-written, but some sections need clarification. I have the following comments and questions:

1. Expression analysis indicated that MOT2.1, MOT2.2 and MOT2.3 genes mainly expressed in different tissues of wild type under different concentrations of molybdate. These results could not be ruled out the functional redundancy of MOT2 genes when one or two of them were knocked out in Arabidopsis. Therefore, authors should generate the double and triple mutants of MOT2 genes, and perform phenotype analysis of these mutants under normal and molybdate deficiency conditions.
2. Since the uptake of molybdenum is impacted by pH value, what is the pH of the growth medium in Figure 4A - 4C?
3. The images in Figure 5 are of poor quality, which makes it had to judge from which the cell types the fluorescent signals are obtained.

Reviewer #2 (Remarks to the Author):

The manuscript of Weber et al. describes Arabidopsis molybdenum transporters of the MOT2 family. The authors found 3 members of this family, one of which has two splice forms. They show that the corresponding proteins are localised in the Golgi and in the plasma membrane and that when expressed in yeast, the proteins transport molybdate. They analysed loss of function mutants but did not observe any effect on growth when grown with or without molybdate. They also showed that the MOT2 proteins interact with Cnx1 involved in MoCo synthesis. In addition, they show evidence for in vitro formation of complex between molybdate and glutathione and speculate on its physiological role. The identification of MOT2 transporters is an important step to better understanding of Mo homeostasis in plants. Therefore, the manuscript describes a highly relevant and important new finding. The conclusions on the identity of MOT2 as molybdate transporters are solid, as is its localization and interaction with Cnx1. However, the physiological relevance of the MOT2 genes has not been addressed sufficiently. The discussion of GSH molybdate complex is an add on not connected to the main story and based purely on in vitro data. The manuscript needs significant attention before it is ready for publication.

1. The physiological role(s) of the MOT2 transporters need to be analysed more comprehensively. The lack of phenotypes of the single mutants points to redundancy, therefore multiple mutants need to be

created and analysed, in particular the triple mutant.

2. Similarly, it would be very informative to analyse double mutants between members of MOT1 and MOT2 family, e.g., *mot1.1* and *mot2.1*, this would give indication about possible other classes of Mo transporters.

3. The manuscript needs careful editing especially for correct naming genes and mutants, often genes are written in small letters, making sometimes difficult to understand.

4. The quantitative data in figures 1, 3, and 4 need to be analysed by two-way ANOVA

5. A diagram describing localization and function of MOT1 and MOT2 would be useful.

Reviewer #3 (Remarks to the Author):

In this manuscript, the authors analyzed members of the MOT2 family for their role in Mo delivery to the sites of its action. Ectopic expression of MOT2.1, MOT2.2 and MOT2.3 in yeast suggests the role in molybdate transport. The author also analyzed tissue specificity of the expression and the subcellular localization of proteins and analyzed the phenotypes of *mot2.1*, *mot2.2*. and *mot2.3* mutants. The authors also showed that MOT2.1, Mot2.3 and Mot2.2A isoform interacts with the molybdenum insertase. All in all, this manuscript increases our understanding of molybdate transport processes in plants.

Comments:

1. The authors use the term "moonlighting" in connection with MOT2 family members' activities even in the title of the manuscript. However, the author do not present any evidence for additional role(s) of MOT2 proteins. Please revise/clarify.

2. Introduction: The presented annotation of molybdate transporters is not clear. The authors mention MOT1.1 and MOT1.2. Is MOT1.2 the same as MOT2 <https://doi.org/10.1111/j.1438-8677.2011.00448.x>? Also, it seems that MOT1 (or MOT1.1.?) was previously categorized as part of the sulfate transporter family. Please clarify accordingly in the introduction to allow the reader to appreciate the motivation for this study. Line 84: I believe the authors meant MOT2 -family function has not been defined.

3. Line 104: "Both proteins share an amino acid similarity..." Please clarify that you are referring to CrMOT1 and AtMOT2.1

4. Results (lines 106-108): the authors present the identification of *mot2.1*- to *mot2.3* as their own finding, while the function of MOT1 and MOT2 family members has been reviewed in Huang et al (<https://doi.org/10.1093/jxb/erab534>) while OsMOT2.1 and OSMOT2.2 were somewhat characterized in rice (DOI: 10.4236/ajps.2013.45A001; PMID: 31797032; PMCID: PMC6305983. These manuscripts should be credited in the result section (section 1) and the Introduction.

5. Fig1A: It will be useful to include Mot2.2B isoform in the analysis.

6. Fig 1C: I would be expecting that time to reach the biomass index should be pretty much the same for - Gal (non-induced conditions), but it follows the pattern for each of the MOT genes. Please comment.

7. Fig. 2A: In the results of MOT2 proteins localization studies, the authors only provided overlaid images. It would be very useful to show the localization of each fluorophore separately as well as to quantify the % of co-localization.

8. Lines 147-149 and Fig. 2 E-L: clarify in the text, which N-or C-terminally GFP-fused MOT2s interacted with apoplast or/and cytosol-targeted GFP segment.

9. Fig. 3H- please provide a higher-resolution image to support the statement that MOT2,3 is expressed in mature ovaries. I would also suggest using Supplemental Figure S2 in the main manuscript as it shows the sites of GUS expression in all tested tissues for all genes.

10. Fig. S3A: StDev and statistical analysis is not presented.
11. It would be useful to establish whether MO2 genes respond transcriptionally to Mo deficiency/excess. While the authors quantified GUS activity in Fig. 3F, they did it only for MOT2.1p-GUS; also, RT-qPCR analysis would be a more accurate test for analyses of the transcriptional response to Mo (and overall, to environmental or developmental perturbations).
12. Lines 211-215: please revise to improve the clarity of statements. Also, in Fig. 4E, it would be useful to use TukeyHSD to identify significant differences between Wt and different mutant lines under different conditions of Mo supply.
13. Please define all abbreviations in the text at the first mention

Cover Letter for Reviewer #1

Dear Reviewer #1,

we would like to thank you for the proper revision of our manuscript and your suggestions to improve our work. We are particularly glad that you appreciated the importance of the topic and hope that our improvements meet with your expectations. A major change in our revised manuscript regarding your suggestions is the addition of data on a *mot1.1 mot2.1* double KO and a *mot1.1. mot1.2 mot2.1* triple KO to meet your demands of higher degree KOs.

Please find our responses to your comments written in blue under each point. We indicated changes in the manuscript using the “track changes” function of MS Word, highlighted the changes in yellow text marker, and specified the lines in which changes were made.

We are looking forward to your response. Please do not hesitate to contact us for further improvements.

With kind regards,

Robert Hänsch, for all authors

Reviewer #1 (Remarks to the Author):

The paper presented a study of the regulatory mechanism of plasma membrane and Golgi-localized MOT2s function in plant molybdate homeostasis. MOT2s was found to interact with the molybdenum-insertase Cnx1 that mediated Moco-biosynthesis proteins complex anchored to actin. This work also revealed GSH was able to bind molybdate via the carboxylate of glycine and the thiol in cysteine to form a complex, which might be responsible for molybdate homeostasis through promoting loading into the vacuole by unspecific HM/GSH transporters. It is concluded that MOT2s is required for molybdate transport in leaves and reproductive organs, thus maintaining molybdate homeostasis in plant tissues. It is therefore important to understand the MOT2 involvement in the regulation of molybdate homeostasis, which plays an important role in SAM-import into Golgi and plant wide molybdate homeostasis. The paper is concise and overall well-written, but some sections need clarification. I have the following comments and questions:

We would like to thank the reviewer for appreciating our work and pointing out the importance to understand plant wide molybdate homeostasis. We are happy to address his comments in the following section to improve our manuscript and to answer remaining questions.

1. Expression analysis indicated that MOT2.1, MOT2.2 and MOT2.3 genes mainly expressed in different tissues of wild type under different concentrations of molybdate. These results could not be ruled out the functional redundancy of MOT2 genes when one or two of them were knocked out in *Arabidopsis*. Therefore, authors should generate the double and triple mutants of MOT2 genes, and perform phenotype analysis of these mutants under normal and molybdate deficiency conditions.

To meet the concern of the reviewer we included phenotype data of higher-degree KOs between the MOT1 and the MOT2-family. In our first manuscript draft we clearly focused on the MOT2-family due to a limited word count. The analyses of higher degree KOs of MOTs expressed in vegetative tissues (*mot1.1*, *mot1.2* and *mot2.1*) is, nevertheless, of interest and we added an additional figure (Fig. S5) to the supplementary analyzing the fresh weight, leaf area, root length, plant survival, and the NR activity of a *mot1.1 / mot2.1* double KO and a *mot1.1 / mot1.2 / mot2.1* triple KO. We changed the counting of supplemental figures in the supplementary and in the main text. We added an according section to the results in chapter 2.5 in ll. 212-218 and 230-239 and discussed the added results in the discussion section in ll. 368-372. We added information regarding the used plant lines to ll. 415-418 and Tab. 1 in the material section and added an additional chapter (chapter 4.8) to the methods section describing the generation of the multi-KO lines by manual crossing of *A. thaliana* plants. We observed that the *mot1.1 mot1.2 mot2.1* triple KO showed a decreased plant survival under molybdate deprivation and absorbed a drastically reduced NR activity under molybdate availability and deprivation conditions. The *mot1.1* single KO serving as control and a *mot1.1. mot2.1* KO showed only slight reactions towards molybdate deprivation. We excluded a *mot1.2* single KO from the analysis since no alterations were observed in a recent study.

We agree with the reviewer that we cannot rule out a certain redundancy between members of the MOT2-family by analyzing single KO lines. We would like to point out that we also did not made a statement in our manuscript regarding this potential redundancy. However, taking the organ-specific expression into account we assume no interchangeability of the MOT2s since no signal of *mot2.2* and *mot2.3* expressions were observed in the vegetative tissue. An important physiological role for MOT2.2 and MOT2.3 in replacing MOT2.1s role during vegetative growth is, thus, unlikely. In our opinion, double and triple KOs of the *mot2*-family would not be beneficial for the statements taken in the manuscript and we would kindly like to argue a meaningful outcome of these experiments. We added a statement in ll. 217-219 to point out that we excluded the analysis of higher degree KO involving *mot2.2* and *mot2.3* on purpose due to different expression patterns.

2. Since the uptake of molybdenum is impacted by pH value, what is the pH of the growth medium in Figure 4A - 4C?

The pH value of the hydroponics growth medium was 5.6 as stated in the cited work of Conn *et al.* 2013. We added this information to the methods section in l. 518.

3. The images in Figure 5 are of poor quality, which makes it hard to judge from which cell types the fluorescent signals are obtained.

We would like to agree with the reviewer in the fact that the fluorescence signal does not allow to tell the cell types from which the image is obtained. To address this concern, we added the information regarding the depicted cell types to the caption of Fig. 5 in l. 271 as it was stated in the methods section. In our opinion not a poor image quality but the character of the fluorescence pattern is remarked by the reviewer. Due to the complexity of the co-expressed fusion constructs necessary for this kind of experiment the resulting fluorescence intensity appears comparably low. We would like to point out that to state protein-protein interaction in this experiment a comparison between the interaction approach and the negative control has to be considered. In Fig. 5 this difference is clearly visible. Detailed images of MOT2 localization are provided in Fig. 2 A-D.

Cover letter for Reviewer #2

Dear Reviewer #2,

we would like to thank you for the detailed revision of our manuscript and the stated improvements of our work. We are happy to hear that you appreciate the relevance of our topic and hope that our improvements meet with your expectations.

A major change to our revised manuscript regarding your comments is the addition of data on a *mot1.1 mot2.1* double KO and a *mot1.1 mot1.2 mot2.1* triple KO as you requested in your comment 2. Furthermore, we analyzed our data using a two-way ANOVA and presented the data in additional supplementary tables.

Please find our responses to your comments written in blue under each point. We indicated changes in the manuscript using the “track changes” function of MS Word, highlighted the changes in yellow text marker, and specified the lines in which changes were made.

We are looking forward to your response. Please do not hesitate to contact us for further improvements.

With kind regards,

Robert Hänsch, for all authors

Reviewer #2 (Remarks to the Author):

The manuscript of Weber *et al.* describes *Arabidopsis* molybdenum transporters of the MOT2 family. The authors found 3 members of this family, one of which has two splice forms. They show that the corresponding proteins are localised in the Golgi and in the plasma membrane and that when expressed in yeast, the proteins transport molybdate. They analysed loss of function mutants but did not observe any effect on growth when grown with or without molybdate. They also showed that the MOT2 proteins interact with Cnx1 involved in MoCo synthesis. In addition, they show evidence for in vitro formation of complex between molybdate and glutathione and speculate on its physiological role. The identification of MOT2 transporters is an important step to better understanding of Mo homeostasis in plants. Therefore, the manuscript describes a highly relevant and important new finding. The conclusions on the identity of MOT2 as molybdate transporters are solid, as is its localization and interaction with Cnx1. However, the physiological relevance of the MOT2 genes has not been addressed sufficiently. The discussion of GSH molybdate complex is an add on not connected

to the main story and based purely on *in vitro* data. The manuscript needs significant attention before it is ready for publication.

We would like to thank the reviewer for the very positive words and pointing out the high relevance of our work. We agree with the reviewer that the *in vitro* data on molybdate-GSH complex formation allows to hypothesize the flow of molybdate in the well-understood mechanisms of GSH heavy metal detoxification in plants. In fact, no specific MOT2 was identified to import molybdate into the vacuole and for the cellular export during senescence. We aimed to deliver a complete model of the cellular molybdate homeostasis taking also the MOT1-family and the GSH-molybdate complex formation into account. We agree with the reviewer that this model needs further proof in future studies since it is based only on *in vitro* data. We are thankful for the comments on our manuscript and have further improved it.

1. The physiological role(s) of the MOT2 transporters need to be analysed more comprehensively. The lack of phenotypes of the single mutants points to redundancy, therefore multiple mutants need to be created and analysed, in particular the triple mutant.

We agree with the reviewer that we cannot rule out a certain redundancy between members of the MOT2-family by analyzing single KO lines. We would like to point out that we also did not made a statement in our manuscript regarding this potential redundancy. However, taking the organ-specific expression into account we assume no interchangeability of the MOT2s since no signal of *mot2.2* and *mot2.3* expressions were observed in the vegetative tissue. An important physiological role for MOT2.2 and MOT2.3 in replacing *Mot2.1s* role during vegetative growth is, thus, unlikely. In our opinion, double and triple KOs of the *mot2*-family would not be beneficial for the statements taken in the manuscript and we would kindly like to argue a meaningful outcome of these experiments. We added a statement in ll. 217-219 to point out that we excluded the analysis of higher degree KO involving *mot2.2* and *mot2.3* on purpose due to different expression patterns. The analysis of higher degree KOs between the *mot1*-family and *mot2.1*, however, appeared to us to be more promising as discussed in our response to the next comment.

2. Similarly, it would be very informative to analyse double mutants between members of MOT1 and MOT2 family, e.g., *mot1.1* and *mot2.1*, this would give indication about possible other classes of Mo transporters.

Due to a limited word count we focused on the members of the MOT2-family in our first draft. We agree with the reviewer that the analyses of higher degree KOs between the MOT1- and the MOT2-family are of interest to identify further mechanisms of molybdate transport. We included data to our manuscript describing the fresh weight, leaf area, root length, plant survival, and the NR activity of a

mot1.1 / mot2.1 double KO and a *mot1.1 / mot1.2 / mot2.1* triple KO in an additional supplementary figure (Fig. S5). We changed the counting of supplemental figures in the supplementary and in the main text. We added an according section to the results in chapter 2.5 in ll. 212-218 and 230-239 and discussed the added results in the discussion section in ll. 368-372. We added information regarding the used plant lines to ll. 415-418 and Tab. 1 in the material section and added an additional chapter (chapter 4.8) to the methods section describing the generation of the multi-KO lines by manual crossing of *A. thaliana* plants. We observed that the *mot1.1 mot1.2 mot2.1* triple KO showed a decreased plant survival under molybdate deprivation and absorbed a drastically reduced NR activity under molybdate availability and deprivation conditions. The *mot1.1* single KO serving as control and a *mot1.1. mot2.1* KO showed only slight reactions towards molybdate deprivation. We excluded a *mot1.2* single KO from the analysis since no alterations were observed in a recent study.

3. The manuscript needs careful editing especially for correct naming genes and mutants, often genes are written in small letters, making sometimes difficult to understand.

In our manuscript we stick to the general nomenclature to write genes in small, italicized letters (*e.g. mot2.1*). Whenever it was necessary to distinguish between different species we added an according abbreviation to the gene written italicized starting with a capital letter (*e.g. Atmot2.1*). Proteins are written in normal letters, all capitalized (MOT2.1). A knockout is described by writing a KO behind the corresponding gene name (*e.g. mot2.1 KO*). In the rare case a gene name is the start of a sentence we capitalized the first letter of the gene for readability. We reviewed our manuscript regarding a consistent way of writing. We will address any requests regarding the correct nomenclature during the further publishing process by the editorial team of Communications Biology.

4. The quantitative data in figures 1, 3, and 4 need to be analysed by two-way ANOVA.

We agree with the reviewer that a two-way ANOVA is a powerful statistical tool to analyze differences in the variance of groups that differ in more than one factor (*e.g. genotype and the presence of molybdate*). However, the Student's T-test is also a well-established statistical method to analyze if the mean of two normally distributed populations differs significantly. Since our statements include only the comparison of two-groups differing in one factor (*genotype or molybdate presence*), in our opinion the Student's T-test is a valid statistical method to underline our statements. To meet the concerns of the reviewer, we performed a two-way ANOVA on the quantitative data and included them in a table in the supplementary (Tab. S2-5). We changed the counting of supplemental tables in the supplementary and in the main text. We analyzed the quantitative data in Fig. 1C using a two-way ANOVA with an Sidak post-hoc test for multiple comparison (Tab. S2). We analyzed the quantitative data in Fig. 3F using a two-way ANOVA using a Tukey post-hoc test for multiple comparison (Tab. S3). We analyzed the data in Fig. 4 D, the molybdate uptake rate, using a one-way ANOVA with a Dunnett's

post-hoc test for multiple comparison (Tab. S4). We analyzed the NR activity in Fig. 4E using a two-way ANOVA with a Tukey post-hoc test for multiple comparison (Tab. S5). We referred to the ANOVA tests in the captions of Fig. 1 (ll. 170-171), Fig. 3 (ll. 204-205), Fig. 4 (ll. 246-247 and 249-250).

5. A diagram describing localization and function of MOT1 and MOT2 would be useful.

We agree with the reviewer that an additional figure describing a model of our findings would be a beneficial addition to our manuscript. A detailed and comprehensive introduction and discussion of the MOT1-family, that would be an integral part of such a figure, would be necessary that does not agree with the limited word count of ~5,000 words of a research article. Due to the large scope of our manuscript using many different methods and providing several experiments, we could not include such a figure. However, we would like to mention that we are currently planning a review article with the goal to compare both, the MOT1- and the MOT2-family as suggested by the reviewer.

Cover letter for Reviewer #3

Dear Reviewer #3,

we would like to thank you for the detailed revision of our manuscript and the stated improvements of our work. We are happy to hear that you appreciate the relevance of our topic and hope that our improvements meet with your expectations.

Please find attached a detailed description of the MOT nomenclature as a response to your comment number 2. As a major change in our revised manuscript we included MOT2.2B to the amino acid sequence analysis in Fig. 1A. Furthermore, we analyzed our data in Fig. 4E using a two-way ANOVA followed by a Tukey post hoc test and presented the data in additional supplementary tables.

Please find our responses to your comments written in blue under each point. We indicated changes in the manuscript using the “track changes” function of MS Word, highlighted the changes in yellow text marker, and specified the lines in which changes were made.

We are looking forward to your response. Please do not hesitate to contact us for further improvements.

With kind regards,

Robert Hänsch, for all authors

Reviewer #3 (Remarks to the Author):

In this manuscript, the authors analyzed members of the MOT2 family for their role in Mo delivery to the sites of its action. Ectopic expression of MOT2.1, MOT2.2 and MOT2.3 in yeast suggests the role in molybdate transport. The author also analyzed tissue specificity of the expression and the subcellular localization of proteins and analyzed the phenotypes of *mot2.1*, *mot2.2*. and *mot2.3* mutants. The authors also showed that MOT2.1, MOT.3 and MOT2.2A isoform interacts with the molybdenum insertase. All in all, this manuscript increases our understanding of molybdate transport processes in plants.

We would like thank the reviewer for appreciating our work and pointing out its relevance. We will thoroughly improve our manuscript regarding the noted concerns and are happy to answer the remaining questions.

Comments:

1. The authors use the term “moonlighting” in connection with MOT2 family members’ activities even in the title of the manuscript. However, the author does not present any evidence for additional roles of MOT2 proteins. Please revise/clarify.

The MOT2 family clearly shows the ability of molybdate transport as shown in our yeast experiments and several members interact with the Moco biosynthesis. In a recent study (Temple *et al.*, 2022), the same MOT2 family was associated with the Golgi-import of S-Adenosyl methionine (SAM), an important agent in cell wall biosynthesis (ll. 88-94). From this dual transport function, we hypothesized a moonlighting character for the MOT2-family in molybdate and SAM transport (ll. 389-398). We discussed this hypothesis with examples of Moco metabolism related proteins that also evolved moonlighting functions due to the ancient character of Moco biosynthesis (ll. 398-403).

2. Introduction: The presented annotation of molybdate transporters is not clear. The authors mention MOT1.1 and MOT1.2. Is MOT1.2 the same as MOT2 <https://doi.org/10.1111/j.1438-8677.2011.00448.x?> Also, it seems that MOT1 (or MOT1.1.?) was previously categorized as part of the sulfate transporter family. Please clarify accordingly in the introduction to allow the reader to appreciate the motivation for this study. Line 84: I believe the authors meant MOT2 -family function has not been defined.

We would like to thank the reviewer for pointing out a problem with correctly identifying the described MOTs. The identification of a second MOT-family by Tejada-Jimenez in 2011 made it necessary to rename the known members of MOT1-family, MOT1 (AGI code AT2G25680) and MOT2 (AGI code AT1G80310). In a study, that we published recently (Minner-Meinen *et al.*, 2022), we clarified the confusing MOT nomenclature by also paying attention to the MOT2 family: MOT1 was renamed to MOT1.1 and MOT2 was renamed to MOT1.2 indicating that both proteins belong to the same family with a high degree of relation. The second, independent family described in the present manuscript was named MOT2-family following the MOT-nomenclature in *C. reinhardtii* established by Tejada-Jimenez *et al.* (2011). In the present manuscript we simplified the MOT nomenclature due to a word count limitation. However, a precise naming of the described MOTs is of interest for us and we are happy for pointing out any ambiguities by the reviewers. To avoid the confusing naming of MOTs we added information to the introduction regarding the former and the current nomenclature (ll. 71 and 75). The AGI code for all MOT2-family members is stated in ll. 107,110, and 111 detail in the manuscript introduction and the methods section in Tab. 1 allowing the clear identification of all involved proteins.

Before proving its character as molybdate transporter, MOT1.1 was categorized as sulfate transporter since molybdate and sulfate share physical-chemical properties like size, charge and their tetrahedral

structure as reviewed in detail in our previous study with focus on the MOT1-family (Minner-Meinen *et al.*, 2022). The present study deals with the independent MOT2-family that is not related to the MOT1-family as we modified in I. 82 We modified the introduction in II. 71-77 to make it easier to distinguish between the MOT1- and MOT2-family and to point our motivation to characterize this new family and to assign their physiological role in *A. thaliana*. We clarified the statement in II. 85-87 accordingly by stating that no molybdate transporter function for the MOT2-family in *Arabidopsis* was proven. The original statement was misleading regarding that not MOT was identified in *Arabidopsis*.

3. Line 104: “Both proteins share an amino acid similarity...” Please clarify that you are referring to CrMOT1 and AtMOT2.1

We clarified the statement in II. 107-108 accordingly in the manuscript.

4. Results (lines 106-108): the authors present the identification of mot2.1- to mot2.3 as their own finding, while the function of MOT1 and MOT2 family members has been reviewed in Huang *et al* (<https://doi.org/10.1093/jxb/erab534>) while OsMOT2.1 and OSMOT2.2 were somewhat characterized in rice (DOI: 10.4236/ajps.2013.45A001; PMID: 31797032; PMCID: PMC6305983. These manuscripts should be credited in the result section (section 1) and the Introduction.

In chapter 2.1 we state that *mot2.1* in *Arabidopsis* was identified by Tejada-Jimenez *et al.* (2011) as cited in I. 107. In fact, we were not aware that Huang *et al.* (2022) identified *Atmot2.2* and *Atmot2.3* using *in silico* analyses. We would like to thank the reviewer for mentioning the Huang *et al.* study allowing us to credit the authors and to give a complete overview on MOT2-research that was done prior to our study. As a consequence of their *in silico* study Huang *et al.* (2022) state that “there are three copies of AtMOT2 in *Arabidopsis* while their molybdate transport activities and functions have not been investigated. Further studies are required to confirm whether MOT2 in higher plants has molybdate transport activity.”

We clearly agree with the reviewer that Huang *et al.* needs to be credited for their *in silico* identification of *Atmot2.2* and *Atmot2.3* and added their citation number 15 to the introduction (I. 68), the first chapter 2.1 in the results section (II. 109-111) and in the beginning of the discussion section (II. 322-323).

The presence of MOT2-family members in *O. sativa* is properly cited from Hibara *et al.* (2013) with citation number 13 in II. 84-85 in the introduction. In the suggested review from Huang *et al.* (2022) the authors share their concerns that the OsMOT2-family is not capable of molybdate transport since no direct transport function was shown. Thus, the authors state that “the molybdate transport activity of MOT2 in higher plants has not been investigated.” Given these concerns, the work of Hibara *et al.* (2013) is credited to a sufficient extend in our opinion.

5. Fig1A: It will be useful to include Mot2.2B isoform in the analysis.

We included MOT2.2B in the analysis in Fig. 1 A.

6. Fig 1C: I would be expecting that time to reach the biomass index should be pretty much the same for – Gal (non-induced conditions), but it follows the pattern for each of the MOT genes. Please comment.

The analyzed MOTs show strikingly different molybdate transport activities. With an Km value of 20 nM as described by Tomatsu *et al.* (2007), MOT1.1 is a super-high affinity transporter, even active under scarce molybdate concentrations. If we now assume that the yeast-derived *gal1* promoter system is not completely silent under non-induced (-Gal) conditions, we have a minimal background expression of each *mot* in the according yeast strain. Due to the different character of each MOT, this background presence has different effects on each strain in the presence of molybdate and chlorate, resulting in a different growth behavior under non-induced (-Gal) conditions.

Furthermore, we would like to point out the duration of our yeast experiments. Each growth curve was recorded for at least 120 h. Here, minimal differences of the initial yeast concentration led to differences at the end of the experiment. Our experimental setup allowed us to use the same yeast stock medium to inoculate induced (+Gal) and non-induced (-Gal) conditions to avoid this effect in each technical replication. However, when using different yeast strains, this normalization was not possible.

We agree with the reviewer that each yeast strain differs in the time to reach the biomass index 40 under non-induced (-Gal) conditions. However, under the aspects of the points discussed above, comparing each yeast strain for its own, definitely allows to validate them as molybdate transporters. In case of the MOT ranking according their transport activity, we would like to point out that we normalized to chlorate free controls (see caption of Fig.2 in ll. 168-170), in which at least the first point mentioned by us was eliminated. This ranking is also discussed with caution in our manuscript.

7. Fig. 2A: In the results of MOT2 proteins localization studies, the authors only provided overlaid images. It would be very useful to show the localization of each fluorophore separately as well as to quantify the % of co-localization.

We added a figure to the supplementary (new Fig. S1) showing each fluorophore separately. We referred to the added supplemental figure in the caption of Fig. 2 in l. 77. We changed the counting of supplemental figures in the supplementary and in the main text.

8. Lines 147-149 and Fig. 2 E-L: clarify in the text, which N-or C-terminally GFP-fused MOT2s interacted with apoplast or/and cytosol—targeted GFP segment.

We clarified our statement in the text in ll. 153-159.

9. Fig. 3H- please provide a higher-resolution image to support the statement that MOT2,3 is expressed in mature ovaries. I would also suggest using Supplemental Figure S2 in the main manuscript as it shows the sites of GUS expression in all tested tissues for all genes.

We changed our statement in the manuscript regarding the resolution of the observed *mot2.3:gus* expression from “in mature ovaries” to the more general term “in ovaries” in ll. 188 and 192. Due to a recommended limit of five figures in the main text we would like to keep Fig. S2 in the supplementary. We would like to point out that the caption of Figure 3 refers to Fig. S2.

10. Fig. S3A: StDev and statistical analysis is not presented.

We modified Fig. S3A and the caption according to the suggestions of the reviewer and analyzed the results using a Student's T-test.

11. It would be useful to establish whether MOT2 genes respond transcriptionally to Mo deficiency/excess. While the authors quantified GUS activity in Fig. 3F, they did it only for MOT2.1p-GUS; also, RT-qPCR analysis would be a more accurate test for analyses of the transcriptional response to Mo (and overall, to environmental or developmental perturbations).

We agree with the reviewer that a qPCR would be an interesting addition to our work. However, we would like to point out that we did not discuss our results in the light of a transcriptional response with the exception stated above. Furthermore, we included no statement regarding environmental or developmental perturbation in our manuscript. Due to the scope of the present manuscript, we would kindly like to argue the necessity of additional qPCR analyses.

The goal of our fluorimetric GUS assay was to compare the *mot2.1*-promoter activity in different organs and tissues. Due to the expression of *mot2.2* and *mot2.3* in only one organ (pollen and ovaries, respectively) a fluorimetric analysis of these promoters would not be a useful addition in our opinion. Also no altered expression patterns were observed under molybdate deficiency.

The histochemical GUS assay allows to analyze the *mot2* expression patterns not only on organ level but also even on tissue level in a detailed way as observed for example in Fig. 3A. Here it was not only able to show an expression in the root but to distinguish between the vascular tissue and the root cortex. In addition, the fluorimetric GUS assay gives quantitative data of the expression strength in the overall organ. This combination allows conclusion about *mot2* expression in a resolution that would not be achievable by using a RT-qPCR.

We discussed the observed substrate induction of the *mot2.1*-promoter quite cautiously since we are aware of the limitations of the method to allow such statements. However, the clarity of the results encouraged us to do so.

12. Lines 211-215: please revise to improve the clarity of statements. Also, in Fig. 4E, it would be useful to use TukeyHSD to identify significant differences between Wt and different mutant lines under different conditions of Mo supply.

We clarified the statements in ll. 227-229 and pointed out that we observed a significant reduction of 30 % under molybdate availability conditions in the *mot2.1* KO compared to the WT. The Student's T-test is a well-established statistical method to analyze if the mean of two normally distributed populations differs significantly. Since our statements include only the comparison of two-groups differing in one factor (genotype or molybdate presence), in our opinion the Student's T-test is a valid statistical method to underline our statements. To meet the concerns of the reviewer, we analyzed the NR activity in Fig. 4E using a two-way ANOVA with a Tukey post-hoc test for multiple comparison and included the results in a table in the supplementary (Tab. S5). We changed the counting of supplemental tables in the supplementary and in the main text.

13. Please define all abbreviations in the text at the first mention

We reviewed our manuscript accordingly.

REVIEWERS' COMMENTS:

Reviewer #1 (Remarks to the Author):

The authors did a good job addressing my concerns and I have no more critical questions about this manuscript.

Reviewer #2 (Remarks to the Author):

In the revised manuscript the authors addressed well all my comments regarding the scientific contents. With respect to the nomenclature, I wonder since when have the Arabidopsis community standards (https://www.arabidopsis.org/links/comm_stan.pdf) been abandoned, but in the end I agree with the authors that this is an editorial issue of the journal.